# A protocol for an intercomparison of biodiversity and ecosystem services models using harmonized land-use and climate scenarios

HyeJin Kim[1,2], Isabel M.D. Rosa[1,2], Rob Alkemade[3,4], Paul Leadley[5], George Hurtt[6], Alexander Popp[7], Detlef P van Vuuren[8], Peter Anthoni[9], Almut Arneth[9], Daniele Baisero[10], Emma Caton[11], Rebecca Chaplin-Kramer[12], Louise Chini[6], Adriana De Palma[11], Fulvio Di Fulvio[13], Moreno Di Marco[14], Felipe Espinoza[11], Simon Ferrier[15], Shinichiro Fujimori[16], Ricardo E. Gonzalez[18], Maya Gueguen[29], Carlos Guerra[1,2], Mike Harfoot[19], Thomas D. Harwood[15], Tomoko Hasegawa[17], Vanessa Haverd[20], Petr Havlík[13], Stefanie Hellweg[21], Samantha L. L. Hill[11,19], Akiko Hirata[22], Andrew J. Hoskins[15], Jan H. Janse[3,23], Walter Jetz[24], Justin A. Johnson[25], Andreas Krause[9], David Leclère[13], Ines S. Martins[1,2], Tetsuya Matsui[22], Cory Merow[24], Michael Obersteiner[13], Haruka Ohashi[22], Benjamin Poulter[26], Andy Purvis[11,27], Benjamin Quesada[9, 28], Carlo Rondinini[10], Aafke M. Schipper[3,29], Richard Sharp[12], Kiyoshi Takahashi[17], Wilfried Thuiller[30], Nicolas Titeux[1,31], Piero Visconti[32,33], Christopher Ware[15], Florian Wolf[1,2], Henrique M. Pereira[1,2,34]

[1]German Centre for Integrative Biodiversity Research (iDiv) Halle-Jena- Leipzig, Deutscher Platz 5e, 04103 Leipzig, Germany
[2]Institute of Biology, Martin Luther University Halle Wittenberg, Am Kirchtor 1, 06108 Halle (Saale), Germany
[3]PBL Netherlands Environmental Assessment Agency, the Hague, the Netherlands
[4]Environmental System Analysis Group, Wageningen University, the Netherlands
[5]Ecologie Systématique Evolution, Univ. Paris-Sud, CNRS, AgroParisTech, Université Paris-Saclay, 91400, Orsay, France
[6]Department of Geographical Sciences, University of Maryland, College Park, MD 20740, USA
[7]Potsdam Institute for Climate Impact Research (PIK), Member of the Leibniz Association, Potsdam, Germany
[8]Copernicus Institute for Sustainable Development, Utrecht University, Utrecht, the Netherlands
[9]Karlsruhe Institute of Technology, Dept. Meteorology and Climate/Atmospheric Environmental Research, Kreuzeckbahnstr. 19, 82467 Garmisch-Partenkirchen, Germany
[10]C/O Global Mammal Assessment program, Department of Biology and Biotechnologies, Sapienza Università di Roma, Viale dell'Univerisità 32, I-00185, Rome, Italy
[11]Department of Life Sciences, Natural History Museum, London SW7 5BD, U.K.
[12]The Natural Capital Project, Stanford University, 371 Serra Mall, Stanford, CA 94305, USA
[13]International Institute for Applied Systems Analysis, Schlossplatz 1, Laxenburg 2361, Austria
[14]CSIRO Land and Water, GPO Box 2583, Brisbane QLD 4001, Australia
[15]CSIRO Land and Water, GPO Box 1700, Canberra ACT 2601, Australia
[16]Kyoto University, Department of Environmental Engineering, 361, C1-3, Kyoto University Katsura Campus, Nishikyo-ku, Kyoto-city, 615-8540 Japan
[17]Center for Social and Environmental Systems Research, National Institute for Environmental Studies (NIES), 16–2 Onogawa, Tsukuba, Ibaraki 305–8506, Japan
[18]Department of Life Sciences, Imperial College London, Silwood Park, Ascot SL5 7PY, U.K.
[19]UN Environment, World Conservation Monitoring Centre, 219 Huntingdon Road, Cambridge, CB3 0DL, UK.
[20]CSIRO Oceans and Atmosphere, Canberra, 2601, Australia
[21]Institute of Environmental Engineering, ETH Zurich, 8093 Zurich, Switzerland
[22]Forestry and Forest Products Research Institute, Forest Research and Management Organization, 1 Matsunosato, Tsukuba, Ibaraki, 305-8687, Japan
[23]Netherlands Inst. of Ecology NIOO-KNAW, Wageningen, the Netherlands
[24]Ecology and Evolutionary Biology, Yale University, 165 Prospect St, New Haven, CT 06511, USA
[25]Institute on the Environment, University of Minnesota, 1954 Buford Ave. St. Paul, MN 55105, USA
[26]NASA GSFC, Biospheric Science Lab., Greenbelt, MD 20771, USA
[27]Department of Life Sciences, Imperial College London, Silwood Park, Ascot SL5 7PY, U.K.

[28]Universidad del Rosario, Faculty of Natural Sciences and Mathematics, Kr 26 No 63B-48, Bogotá D.C, Colombia

[29]Institute for Water and Wetland Research, PO Box 9010, 6500 GL Nijmegen, the Netherlands

[30]Univ. Grenoble Alpes, CNRS, Univ. Savoie Mont Blanc, Laboratoire d'Écologie Alpine (LECA), F-38000 Grenoble, France

[31]Helmholtz Centre for Environmental Research - UFZ, Department of Community Ecology, Theodor-Lieser-Strasse 4, D-06210 Halle, Germany.

[32]Institute of Zoology, Zoological Society of London, Regent's Park, London, NW1 4RY

[33]Centre for Biodiversity and Environment Research, University College London, Gower Street, London, C1E6BT

[34]CIBIO/InBIO, Centro de Investigação em Biodiversidade e Recursos Genéticos, Cátedra REFER-Biodiveridade,

Universidade do Porto, Campus Agrário de Vairão, R. Padre Armando Quintas, 4485-661 Vairão, Portugal

*Correspondence to*: Henrique M. Pereira (hpereira@idiv.de)

**Abstract.** To support the assessments of the Intergovernmental Science-Policy Platform on Biodiversity and Ecosystem Services (IPBES), the IPBES Expert Group on Scenarios and Models is carrying out an intercomparison of biodiversity and

ecosystem services models using harmonized scenarios (BES-SIM). The goals of BES-SIM are (1) to project the global impacts of land use and climate change on biodiversity and ecosystem services (i.e., nature's contributions to people) over the coming decades, compared to the 20th century, using a set of common metrics at multiple scales, and (2) to identify model uncertainties and research gaps through the comparisons of projected biodiversity and ecosystem services across models. BES-SIM uses three scenarios combining specific Shared Socio-economic Pathways (SSPs) and Representative Concentration Pathways

(RCPs) – SSP1xRCP2.6, SSP3xRCP6.0, SSP5xRCP8.6 – to explore a wide range of land-use change and climate change futures. This paper describes the rationale for scenarios selection, the process of harmonizing input data for land use, based on the second phase of the Land Use Harmonization Project (LUH2), and climate, the biodiversity and ecosystem service models used, the core simulations carried out, the harmonization of the model output metrics, and the treatment of uncertainty. The results of this collaborative modelling project will support the ongoing global assessment of IPBES, strengthen ties between

IPBES and the Intergovernmental Panel on Climate Change (IPCC) scenarios and modelling processes, advise the Convention on Biological Diversity (CBD) on its development of a post-2020 strategic plans and conservation goals, and inform the development of a new generation of nature-centred scenarios.

## 1 Introduction

Understanding how anthropogenic activities impact biodiversity and human societies is essential for nature conservation and

sustainable development. Land use and climate change are widely recognized as two of the main drivers of future biodiversity change (Hirsch and CBD, 2010; Maxwell et al., 2016; Sala, 2000; CBD and UNEP, 2014) with potentially severe impacts on ecosystem services and ultimately human well-being (Cardinale et al., 2012; MA, 2005). Habitat and land-use changes, resulting from past, present and future human activities, as well as climate change, have both immediate and long term impacts on biodiversity and ecosystem services (Graham et al., 2017; Lehsten et al., 2015; Welbergen et al., 2008). Therefore, current

and future land-use projections are essential elements for assessing biodiversity and ecosystem change (Titeux et al., 2016,

2017). Climate change has already observed to have direct and indirect impact on biodiversity and ecosystems and it is projected to intensify as we approach the end of the century with potentially severe consequences on species and habitats, thereby also on ecosystem functions and ecosystem services (Pecl et al., 2017; Settele et al., 2015).

Global environmental assessments, such as the Millennium Ecosystem Assessment (MA 2005), the Global Biodiversity Outlooks (GBO), the multiple iterations of the Global Environmental Outlook (GEO), the Intergovernmental Panel on Climate Change (IPCC), and other studies have used scenarios to assess the impact of socio-economic development pathways on land use and climate and their consequences for biodiversity and ecosystem services (Jantz et al., 2015; Pereira et al., 2010). Models are used to quantify the biodiversity and ecosystem services impacts of different scenarios, based on climate and land-use projections from General Circulation Models (GCM) and Integrated Assessment Models (IAM) (Pereira et al., 2010). These models include empirical dose-response models, species-area relationship models, species distribution models and more mechanistic models such as trophic ecosystem models (Pereira et al., 2010; Akçakaya et al., 2016). So far, each of these scenario exercises has been based on a single model or a small number of biodiversity and ecosystem service models, and intermodel comparison and uncertainty analysis have been limited (IPBES, 2016; Leadley et al., 2014). The Expert Group on Scenarios and Models of the Intergovernmental Science-Policy Platform on Biodiversity and Ecosystem Services (IPBES) is addressing this gap by carrying out a biodiversity and ecosystem services model intercomparison with harmonized scenarios, for which this paper lays out the protocol.

Over the last two decades, IPCC has fostered the development of global scenarios to inform climate mitigation and adaptation policies. The Representative Concentration Pathways (RCPs) describe different climate futures based on greenhouse gas emissions over the 21st century (van Vuuren et al., 2011). These emissions pathways have been converted into climate projections in the most recent Climate Model Inter-comparison Project (CMIP5). In parallel, the climate research community also developed the Shared Socio-economic Pathways (SSPs), which consist of trajectories of future human development with different socio-economic conditions and associated land-use projections (Popp et al., 2017; Riahi et al., 2017). The SSPs can be combined with RCP-based climate projections to explore a range of futures for climate change and land-use change and are being used in a wide range of impact modelling intercomparisons (Rosenzweig et al., 2017; van Vuuren et al., 2014). Therefore, the use of the SSP-RCP framework for modelling the impacts on biodiversity and ecosystem services provides an outstanding opportunity to build bridges between the climate, biodiversity and ecosystem services communities, and has been explicitly recommended as a research priority in the IPBES assessment on scenarios and models (IPBES, 2016).

Model intercomparisons bring together different communities of practice for comparable and complementary modelling, in order to improve the comprehensiveness of the subject modelled, and to estimate uncertainties associated with scenarios and models (Frieler et al., 2015). In the last decades, various model intercomparison projects (MIPs) have been initiated to assess the magnitude and uncertainty of climate change impacts. For instance, the Inter-Sectoral Impact Model Intercomparison Project (ISIMIP) was initiated in 2012 to quantify and synthesize climate change impacts across sectors and scales (Rosenzweig et al., 2017; Warszawski et al., 2014). The ISIMIP aims to bridge sectors such as agriculture, forestry,

fisheries, water, energy, and health with Global Circulation Models , Earth System Models (ESMs), and Integrated Assessment Models for more integrated and impact-driven modelling and assessment (Frieler et al., 2017).

Here, we present the methodology used to carry out a BES-SIM in terrestrial and freshwater ecosystems. The BES-SIM project addresses the following questions: (1) What are the projected magnitudes and spatial distribution of biodiversity and ecosystem services under a range of climate and land-use future scenarios? (2) What is the magnitude of the uncertainties associated with the projections obtained from different models and scenarios? Whereas independent of the ISI-MIP, the BES-SIM has been inspired by ISI-MIP and other intercomparison projects and was delivered to address the needs of the global assessment of IPBES. We brought together ten biodiversity models and six ecosystem functions and ecosystem services models to assess impacts of land-use and climate change scenarios in coming decades (up to 2070) and to hindcast changes to the last century (to 1900). The modelling approaches differ in several ways in how they treat biodiversity and ecosystem services responses to land use and climate changes, including the use of correlative, deductive, and process-based approaches, and in how they treat spatial scale and temporal dynamics. We assessed different classes of Essential Biodiversity Variables (EBV) including species populations, community composition and ecosystem function, as well as a range of measures on ecosystem services such as food production, pollination, water quantity and quality, climate regulation, soil protection, and pest control (Pereira et al. 2010; Akçakaya et al., 2016). This paper provides an overview of the scenarios, models and metrics used in this intercomparison, thus a roadmap for further analyses that is envisaged to be integrated into the first global assessment of the IPBES (Figure 1).

## 2 Scenarios selection

All the models involved in BES-SIM used the same set of scenarios using particular combinations of SSPs and RCPs. In the selection of the scenarios, we used the following criteria: 1) data on projections should be readily available, and 2) the total set should cover a broad range of land-use change and climate change projections. The first criterion implied that we selected SSP-RCP combinations that are included in the ScenarioMIP protocol as part of CMIP6 (O'Neill et al., 2016), as harmonised data was available for these runs and these form the basis of the CMIP climate simulations. The second criteria implied a selection within the ScenarioMIP set of scenarios with low and high degrees of climate change and different land-use scenarios. Our final selection was SSP1 with RCP2.6 (moderate land-use pressure and low level of climate change) (van Vuuren et al., 2017), SSP3 with RCP6.0 (high land-use pressure and moderately high level of climate change) (Fujimori et al., 2017), and SSP5 with RCP8.5 (medium land-use pressure and very high level of climate change) (Kriegler et al., 2017), thus allowing us to assess a broad range of plausible futures (Table 1). Further, by combining projections of low and high anthropogenic pressure of land-use with low and high levels of climate change projections, we can test these drivers' individual and synergistic impacts on biodiversity and ecosystem services.

The first scenario (SSP1xRCP2.6) is characterized by relatively "environmentally-friendly world" with low population growth, high urbanization, relatively low demand for animal products and high agricultural productivity. These factors together

lead to a decrease in the land use of around 700 Mha globally over time (mostly pastures). This scenario is also characterised by low air pollution, while policies are introduced to limit the increase of greenhouse gases in the atmosphere, leading to an additional forcing of 2.6 W/m$^2$ before 2100. The second scenario (SSP3xRCP6.0) is characterised by "regional rivalry", with high population growth, slow economic development, material-intensive consumption and low food demand per capita. Agricultural land intensification is low, especially due to the very limited transfer of new agricultural technologies to developing countries. This scenario has land-use change hardly regulated, with a large land conversion for human-dominated uses, and has a relatively high level of climate change with a radiative forcing of 6.0 W/m$^2$ by 2100. The third scenario (SSP5xRCP8.5) is a world characterised by "strong economic growth" fuelled by fossil fuels, with low population growth, high urbanization, high food demand per capita but also high agricultural productivity. As a result, there is a modest increase in land use. Air pollution policies are stringent, motivated by local health concerns. This scenario leads to a very high level of climate change with a radiative forcing of 8.5 W/m$^2$ by 2100. Full descriptions of each SSP scenario are given in Popp et al. (2017) and Riahi et al. (2017). The SSP scenarios excluded elements that have interaction effects with climate change except for SSP1, which focuses on environmental sustainability. Thus, SSPs describe futures where biodiversity is not affected by climate change to allow for the important estimation of the climate change impact on biodiversity (O'Neill et al., 2014).

## 3 Input data

A consistent set of land use and climate data was used across the models to the extent possible, using existing datasets. All models in BES-SIM used the newly released Land Use Harmonization dataset version 2 (LUH2, Hurtt et al., 2018). For the models that require climate data, we selected the climate projections of the past, present and future from CMIP5 / ISIMIP2a (McSweeney and Jones, 2016) and its downscaled version from the WorldClim (Fick and Hijmans, 2017), as well as MAGICC 6.0 (Meinshausen et al., 2011a, 2011b) from the IMAGE model for GLOBIO models (Table 2). A complete list of input datasets and variables used by the models is documented in Table S1 of the Supplement.

### 3.1 Land cover and land-use change data

The land-use scenarios provide an assessment of land-use dynamics in response to a range of socio-economic drivers and their consequences for the land system. The IAMs used to model land-use scenarios – IMAGE for SSP1/RCP2.6, AIM for SSP3/RCP7.0, and REMIND/MAgPIE for SSP5/RCP8.5 – include different economic and land-use modules for the translation of narratives into consistent quantitative projections across scenarios (Popp et al., 2017). It is important to note that the land-use scenarios used, although driven mostly by the SSP storylines, were projected to be consistent with the paired RCPs and include biofuel deployment to mitigate climate change. The SSP3 is associated with RCP7.0 (SSP3xRCP7.0); however, climate projections (i.e., time series of precipitation and temperature) are currently not available for RCP7.0. Therefore, we chose the closest RCP available, which was RCP6.0, for the standalone use of climate projections and chose SSP3xRCP6.0 for the land use projections from the LUH2. In this paper, we refer to this scenario as SSP3xRCP6.0.

The land-use projections from each of the IAMs were harmonized using the LUH2 methodology. LUH2 was developed for CMIP6 and provides a global gridded land-use dataset comprising estimates of historical land-use change (850-2015) and future projections (2015-2100), obtained by integrating and harmonizing land-use history with future projections of different IAMs (Jungclaus et al., 2017; Lawrence et al., 2016; O'Neill et al., 2016). Compared to the first version of the LUH (Hurtt et al., 2011), LUH2 (Hurtt et al., 2018) is driven by the latest SSPs, has a higher spatial resolution (0.25 vs 0.50 degree), more detailed land-use transitions (12 versus 5 possible land-use states), and increased data-driven constraints (Heinimann et al., 2017; Monfreda et al., 2008). LUH2 provides over 100 possible transitions per grid cell per year (e.g., crop rotations, shifting cultivation, agricultural changes, wood harvest) and various agricultural management layers (e.g., irrigation, synthetic nitrogen fertilizer, biofuel crops), all with annual time steps. The 12 states of land include the separation of primary and secondary natural vegetation into the forest and non-forest sub-types, pasture into managed pasture and rangeland, and cropland into multiple crop functional types (C3 annual, C3 perennial, C4 annual, C4 perennial, and N fixing crops) (Table 3).

For biodiversity and ecosystem services models that rely on discrete, high-resolution land-use data (i.e., the GLOBIO model for terrestrial biodiversity and the InVEST model), the fractional LUH2 data were downscaled to discrete land-use grids (10 arc-seconds resolution; ~300 m) with the land-use allocation routine of the GLOBIO4 model. To that end, the areas of urban, cropland, pasture, rangeland and forestry from LUH2 were first aggregated across the LUH2 grid cells to the regional level of the IMAGE model, with forestry consisting of the wood harvest from forested cells and non-forested cells with primary vegetation. Next, the totals per region were allocated to 300m cells with the GLOBIO4 land allocation routine, with specific suitability layers for urban, cropland, pasture, rangeland, and forestry. After allocation, cropland was reclassified into three intensity classes (low, medium, high) based on the amount of fertilizer per grid cell. More details on the downscaling procedure are provided in Supplementary Methods in the Supplement.

### 3.2 Climate data

General Circulation Models are based on fundamental physical processes (e.g., conservation of energy, mass, and momentum and their interaction with the climate system) and simulate climate patterns of temperature, precipitation, and extreme events at a large scale (Frischknecht et al., 2016). Some GCMs now incorporate elements of Earth's climate system (e.g., atmospheric chemistry, soil and vegetation, land and sea ice, carbon cycle) in Earth Systems Models (GCM with interactive carbon cycle), and have dynamically downscaled models with higher resolution data in Regional Climate Models (RCMs).

A large number of climate datasets are available today from multiple GCMs, but not all GCMs provide projections for all RCPs. In BES-SIM, some models require continuous time-series data. In order to harmonize the climate data to be used across biodiversity and ecosystem service models, we chose the bias-corrected climate projections from CMIP5, which were also adopted by ISIMIP2a (Hempel et al., 2013) or their downscaled versions available from WorldClim (Fick and Hijmans, 2017). Most analyses were carried out using a single GCM, the IPSL-CM5A-LR (Dufresne et al., 2013), since it provides mid-range projections across the five GCMs (HadGEM2-ESGFDL-ESM2M, IPSL-CM5A-LR, MIROC-ESM-CHEM, and NorESM1-M) in ISIMIP2a (Warszawski et al., 2014).

The ISIMIP2a output from the IPSL-CM5A-LR provides 12 climate variables on daily time steps from the pre-industrial period 1951 to 2099 at 0.5-degree resolution (McSweeney and Jones, 2016), of which only a subset was used in this exercise (Table S1). The WorldClim downscaled dataset has 19 bioclimatic variables derived from monthly temperature and rainfall from 1960 to 1990 with multi-year averages for specific points in time (e.g., 2050, 2070) up to 2070. Six models in BES-SIM used ISIMIP2a dataset and three models used WorldClim. An exception was made to the GLOBIO models, which used MAGICC 6.0 climate data (Meinshausen et al., 2011b, 2011a) in the IMAGE model framework (Stehfest et al., 2014), to which GLOBIO is tightly connected (Table 2). The variables used from climate dataset in each model are listed in Table S1.

### 3.3 Other input data

In addition to the land-use and climate data, most models use additional input data to run their future and past simulations to estimate changes in biodiversity and ecosystem services. For instance, species occurrence data are an integral part of modelling in six of ten biodiversity models while two models rely on estimates of habitat affinity coefficients (e.g., reductions in species richness in a modified habitat relative to the pristine habitat) from the PREDICTS model (Newbold et al., 2016; Purvis et al., 2018). In three Dynamic Global Vegetation Models (DGVM) models, atmospheric $CO_2$ concentrations, irrigated fraction, and wood harvest estimates are commonly used, while two ecosystem services models rely on topography and soil type data for soil erosion measures. A full list of model-specific input data is listed in Table S1.

### 4 Models in BES-SIM

Biodiversity and ecosystem services models at the global scale have increased in number and improved considerably over the last decade, especially with the availability of biodiversity data and advancement in statistical modelling tools and methods (IPBES, 2016). In order for a model to be included in BES-SIM, it had either to be published in a peer-reviewed journal or adopt published methodologies, with modifications made to modelling sufficiently documented and accessible for review (Table S2). Sixteen models participated in BES-SIM (Appendix 1, details on modelling methods in Table S2). These models were mainly grouped into four classes: species-based, community-based, and ecosystem-based models of biodiversity, and models of ecosystem functions and services. The methodological approaches, the taxonomic or functional groups, the spatial resolution and the output metrics differ across models (Appendix 1). All sixteen models are spatially explicit with 15 of them using land-use data as an input, 13 of them requiring climate data. We also used one model, BIOMOD2 (Thuiller, 2004; Thuiller et al., 2009), to assess the uncertainty of climate range projections without the use of land-use data.

### 4.1 Species-based models of biodiversity

Species-based models aim to predict historical, current, and future potential distribution and abundance of individual species. These can be developed using correlative methods based on species observation and environmental data (Aguirre-Gutiérrez et

al., 2013; Guisan and Thuiller, 2005; Guisan and Zimmermann, 2000), as well as expert-based solutions where data limitations exist (Rondinini et al., 2011). Depending on the methodologies employed and the ecological aspects modelled, they can be known as species distribution models, ecological niche models, bioclimatic envelop models and habitat suitability models (Elith and Leathwick, 2009), and they have been used to forecast environmental impacts on species distribution and status.

In BES-SIM, four species-based models were included: AIM-biodiversity (Ohashi et al., submitted), InSiGHTS (Rondinini et al., 2011; Visconti et al., 2016), MOL (Jetz et al., 2007; Merow et al., 2013), and BIOMOD2 (Appendix 1, Table S2). The first three models project individual species distributions across a large number of species by combining projections of climate impacts on species ranges with projections of land-use impacts on species ranges. AIM-biodiversity uses Global Biodiversity Information Facility (GBIF) species occurrence data of 9,025 species in five taxonomic groups (amphibians, birds,

mammals, plants, reptiles) to train statistical models for current land use and climate to project future species distributions. InSiGHTS uses species' presence records from regular sampling within species' ranges and pseudo-absence records from regular sampling outside of species' ranges on 2,827 species of mammals. MOL uses species land cover preference information and species presence and absence predictions on 20,833 species of amphibians, birds and mammals. Both models rely on IUCN's expert-based range maps as a baseline, which are developed based on expert knowledge of the species habitat

preferences and areas known to be absent (Fourcade, 2016). InSiGHTS and MOL used a hierarchical approach with two steps: first, a statistical model trained on current species ranges is used to assess future climate suitability within species ranges; second, a model detailing associations between species and habitat types based on expert opinion is used to assess the impacts of land-use in the climate suitable portion of the species range. BIOMOD2 is an R modelling package that runs up to nine different algorithms (e.g., random forests, logistic regression) of species distribution models using the same data and the same

framework. BIOMOD2 included three taxonomic groups (amphibians, birds, mammals) (see section 7. Uncertainties).

### 4.2 Community-based models of biodiversity

Community-based models predict the assemblage of species using environmental data and assess changes in community composition through species presence and abundance (D'Amen et al., 2017). Output variables of community-based models include assemblage-level metrics such as the proportion of species persisting in a landscape, mean species abundances (number

of individuals per species), and compositional similarity (pairwise comparison at the species level) relative to a baseline (typically corresponding to a pristine landscape).

   Three models in BES-SIM – cSAR-iDiv (Martins and Pereira, 2017), cSAR-IIASA-ETH (Chaudhary et al., 2015), BILBI (Hoskins et al., in prep.; Ferrier et al., 2004, 2007) – rely on versions of the species-area relationship (SAR) to estimate the proportion of species persisting in human-modified habitats relative to native habitat (i.e., number of species in modified

landscape divided by number of species in the native habitat). In its classical form, the SAR describes the relationship between the area of native habitat and the number of species found within that area. The countryside SAR (cSAR) builds on the classic SAR but accounts for the differential use of both human-modified and native habitats by different functional species groups. Both the cSAR-iDiv and the cSAR-IIASA-ETH models use habitat affinities (proportion of area of a habitat type that can be

effectively used by a species group) to weight the areas of the different habitats in a landscape. The habitat affinities are calibrated from field studies by calculating the change in species richness in a modified habitat relative to the native habitat. The habitat affinities of the cSAR-iDiv model are estimated from the PREDICTS dataset (Hudson et al. 2017; Hudson et al. 2016) while the habitat affinities of the cSAR-IIASA-ETH come from a previously published database of studies (Chaudhary et al., 2015). The cSAR-iDiv model considers 9,853 species for one taxonomic group (birds) in two functional groups (forest species and non-forest species) while the cSAR-IIASA-ETH considers a total of 1,911,583 species for five taxonomic groups (amphibians, birds, mammals, plants, and reptiles) by ecoregions (these are however not 1,911,583 unique species as a species present in two ecoregions will be counted twice). BILBI couples application of the species-area relationship with correlative statistical modelling of continuous patterns of spatial turnover in the species composition of communities as a function of environmental variation. Through space-for-time projection of compositional turnover (i.e., change in species), this coupled model enables the effects of both climate change and habitat modification to be considered in estimating the proportion of species persisting for 254,145 vascular plant species globally.

Three community-based models – PREDICTS, GLOBIO Aquatic and GLOBIO Terrestrial (Alkemade et al., 2009; Janse et al., 2015; Schipper et al., 2016) – estimate a range of assemblage-level metrics based on empirical dose-response relationships between pressure variables (e.g., land-use change and climate change) and biodiversity variables (e.g., species richness or mean species abundance) (Appendix 1). PREDICTS uses a hierarchical mixed-effects model to assess how a range of site-level biodiversity metrics respond to land use and related pressures, using a global database of 767 studies, including over 32,000 sites and 51,000 species in a wide range of taxonomic groups (Hudson et al. 2017; Hudson et al. 2016). GLOBIO is an integrative modelling framework for aquatic and terrestrial biodiversity that builds upon correlative relationships between biodiversity intactness and pressure variables, established with meta-analyses of biodiversity data retrieved from the literature on a wide range of taxonomic groups.

**4.3 Ecosystem-based model of biodiversity**

The Madingley model (Harfoot et al., 2014b) is a mechanistic individual-based model of ecosystem structure and function. It encodes a set of fundamental ecological principles to model how individual heterotrophic organisms with a body size greater than 10 µg that feed on other living organisms interact with each other and with their environment. The model is general in the sense that it applies the same set of principles for any ecosystem to which it is applied, and is applicable across scales from local to global. To capture the ecology of all organisms, the model adopts a functional trait-based approach with organisms characterised by a set of categorical traits (feeding mode, metabolic pathway, reproductive strategy and movement ability), as well as continuous traits (juvenile, adult and current body mass). Properties of ecological communities emerge from the interactions between organisms, influenced by their environment. The functional diversity of these ecological communities can be calculated as well as the dissimilarity over space or time between communities (Table S2). Madingley uses three functional groups (trophic levels, metabolic pathways, reproductive strategies).

## 4.4 Models of ecosystem functions and services

In order to measure ecosystem functions and services, three DGVM models – LPJ-GUESS (Lindeskog et al., 2013; Olin et al., 2015; Smith et al., 2014), LPJ (Poulter et al., 2011; Sitch et al., 2003), CABLE (Haverd et al., 2017) – and three ecosystem services models – InVEST (Sharp et al., 2014), GLOBIO (Alkemade et al., 2009, 2014; Schulp et al., 2012), GLOSP (Guerra

et al., 2016)) – were engaged in this model intercomparison. The DGVMs are process-based models that simulate responses of potential natural vegetation and associated biogeochemical and hydrological cycles to changes in climate and atmospheric $CO_2$ and disturbance regime (Prentice et al., 2007). Processes in anthropogenically managed land (crop, pasture and managed forests) are also increasingly being accounted for (Arneth et al., 2017). DGVMs can project changes in future ecosystem state (e.g., type of plant functional trait (PFT), relative distribution of each PFT, biomass, height, leaf area index, water stress),

ecosystem functioning (e.g., moderation of climate, processing/filtering of waste and toxicants, provision of food and medicines, modulation of productivity, decomposition, biogeochemical and nutrient flows, energy, matter, water), and habitat structure (i.e., amount, composition and arrangement of physical matter that describe an ecosystem within a defined location and time); however, DGVMs are limited in capturing species-level biodiversity change because vegetation is represented by a small number of plant functional types (PFTs) (Bellard et al., 2012; Thuiller et al., 2013).

The InVEST suite includes 18 models that map and measure the flow and value of ecosystem goods and services across a land or a seascape. They are based on biophysical processes of the structure and function of ecosystems and accounts for both supply and demand. The GLOBIO model estimates ecosystem services based on outputs from the IMAGE model (Stehfest et al., 2014), the global hydrological model PCRaster Global Water Balance (PCR-GLOBWB, van Beek et al., 2011), and the Global Nutrient Model (Beusen et al., 2015). It is based on correlative relationships between ecosystem functions and services

and particular environmental variables (mainly land use), quantified based on literature data. Finally, GLOSP is a 2D model that estimates the level of global and local soil erosion and protection using the Universal Soil Loss Equation.

## 5 Output metrics

Given the diversity of modelling approaches, a wide range of biodiversity and ecosystem services metrics can be produced by the model set (Table S2). For the biodiversity model intercomparison analysis, three main categories of common output metrics

were reported over time: extinctions as absolute change in species richness (N, number of species) or as proportional species richness change (P, % species); abundance-based intactness (I, % intactness); and mean proportional change in suitable habitat extent across species (H, % suitable habitat) (Table 4). These metrics were calculated at two scales: local or grid cell (α scale, i.e. the value of the metric within the smallest spatial unit of BES-SIM which is the grid cell) and regional or global (γ scale, i.e. the value of the metric for a set of grid cells comprising a region). For species richness change, some models project the α

metrics at the level of the grid cell (e.g., species-based and SAR based community models) while others average the local point values of the metrics across the grid cell weighted by the area of the different habitats in the cell (e.g., PREDICTS, GLOBIO). In addition, some models only provided α values while others provided both α and γ values (Table 4). For the models that can

project γ metrics, both regional-γ for each IPBES regions (Table 1 in Brooks et al., 2016, UNEP-WCMC, 2015) and a global-γ were reported.

The species diversity change metrics measured as absolute number or percentage change in species richness shows species persistence and extinction in given time and place. Absolute changes in species richness and proportional species richness change are interrelated and may be calculated from reporting species richness over time, as $N_t = S_t - S_{t0}$ and $P = N_t / S_{t0}$, where $S_t$ is the number of species at time $t$. Most models reported one or both types of species richness metrics (Table 4). The abundance-based intactness (I) measures the mean species abundance in the current community relative to the abundances in a pristine community. This metric is available only for two community-based models, i.e., GLOBIO (where intactness is estimated as the arithmetic mean of the abundance ratios of the individual species, whereby ratios >1 are set to 1), and PREDICTS (where intactness is estimated as the ratios of the sum of species abundances). The habitat change (H) measures cell-wise changes in available habitat for the species. It is the changes in the suitable habitat extent of each species relative to a baseline, i.e., $(E_{i,t} - E_{i,t0})/E_{i,t0}$, where $E_{i,t}$ is the suitable habitat extent of species $i$ at time $t$ within the unit of analysis. It is reported by averaging across species occurring in each unit of analysis (grid cell, region, or globe), and is provided by the species-level models (i.e., AIM-biodiversity, InSiGHTS, MOL) (Table 4). The baseline year, $t_0$, used to calculate changes for the extinction and habitat extent metrics, was the first year of the simulation (in most cases $t_0$=1900, see Table 5).

For ecosystem functions and services, each model's output metrics were mapped onto the new classification of Nature's Contributions to People (NCP) published by the IPBES scientific community (Díaz et al., 2018). Among the 18 possible NCPs, the combination of models participating in BES-SIM were able to provide measures for 10 NCPs, including regulating metrics on pollination (e.g., proportion +of agricultural lands whose pollination needs are met, % agricultural area), climate (e.g., vegetation carbon, total carbon uptake and loss, MgC), water quantity (e.g., monthly runoff, Pg/month), water quality (e.g., nitrogen and phosphorus leaching, PgN/s), soil protection (e.g., erosion protection, 0-100 index), hazards (e.g., costal vulnerability, unitless score; flood risk, number of people affected) and detrimental organisms (e.g., fraction of cropland potentially protected by the natural pest relative to all available cropland, km$^2$), and material metrics on bioenergy (e.g., bioenergy-crop production, PgC/yr), food and feed (e.g., total crop production, $10^9$KCal) and materials (e.g., wood harvest, KgC) (Table 6). Some of these metrics require careful interpretation in the context of NCPs (e.g., an increase in flood risk can be caused by climate change and/or by a reduction of the capacity of ecosystems to reduce flood risk) and additional translation of increasing or declining measures of ecosystem functions and services (e.g., food and feed, water quantity) into contextually relevant information (i.e., positive or negative impacts) on human well-being and quality of life. Given the disparity of metrics across models within each NCP category, names of the metrics are listed in Table 6, and units, definitions and methods are provided in Table S3.

## 6 Core simulations

The simulations for BES-SIM required a minimum of two outputs from the modelling teams: present (2015) and future (2050). Additionally, a past projection (1900) and a further future projection (2070) were also provided by several modelling teams. Some models projected further into the past and also at multiple time points from the past to the future (Appendix 1). Models that simulated a continuous time-series of climate change impacts provided 20-year averages around these mid-points to account for inter-annual variability. The models ran simulations at their original spatial resolutions (Appendix 1), and upscaled results to one-degree grid cells using arithmetic means. In order to provide global or regional averages of the α or grid cell metrics, the arithmetic mean values across the cells of the globe or a region were calculated, as well as percentiles of those metrics. Both, one-degree rasters and a table with values for each IPBES region (Table 1 in Brooks et al., 2016, UNEP-WCMC, 2015) and the globe were provided by each modelling team for each output metric.

To measure the individual and synergistic impacts of land use and climate change on biodiversity and ecosystem services, models accounting for both types of drivers were run three times: with land-use change only, with climate change only, and with both drivers combined. For instance, to measure the impact of land use alone, the projections into 2050 were obtained while retaining climate data constant from the present (2015) to the future (2050). Similarly, to measure the impact of climate change alone, the climate projections into 2050 (or 2070) were obtained while retaining the land-use data constant from the present (2015) to the future (2050). Finally, to measure the impact of land use and climate change combined, models were run using projections of both land use and climate change into 2050 (or 2070). When backcasting to 1900, for the models that required continuous climate time-series, random years in the period 1951 to 1960 from the ISIMIP 2a IPSL climate dataset were used to fill the data missing for years 1901 to 1950. Models that used multi-decadal climate averages (i.e., InSiGHTS, BILBI) assumed no climate impacts for 1900.

## 7 Uncertainties

Reporting uncertainty is a critical component of model intercomparison exercises (IPBES, 2016). Within BES-SIM, uncertainties were explored by each model reporting the mean values of its metrics, and where possible the 25th, 50th, and 75th percentiles based on the parameterizations set specific to each model, which can be found in each model's key manuscripts describing the modelling methods; and when combining the data provided by the different models, the average and the standard deviation of the common metrics were calculated (e.g., intermodel average and standard deviation of Pγ). In a parallel exercise to inform BES-SIM, the BIOMOD2 model was used in assessing the uncertainty in modelling changes in species ranges arising from using different RCP scenarios, different GCMs, a suite of species distribution modelling algorithms (e.g., random forest, logistic regression) and different species dispersal hypotheses.

**8 Conclusion**

The existing SSP and RCP scenarios provided a consistent set of past and future projections of two major drivers of terrestrial and freshwater biodiversity change – land use and climate. However, we acknowledge that these projections have certain limitations. These include limited inclusion of biodiversity-specific policies in the storylines (only the SSP1 baseline emphasises additional biodiversity policies) (O'Neill et al., 2016; Rosa et al., 2017), coarse spatial resolution, and land-use classes that are not sufficiently detailed to fully capture the response of biodiversity to land-use change (Harfoot et al., 2014a; Titeux et al., 2016, 2017). The heterogeneity of models and their methodological approaches, as well as additional harmonization of metrics of ecosystem functions and services (Tables 6, S3) are areas for further work. In the future, it will be also important to capture the uncertainties associated with input data, with a focus on uncertainty in land-use and climate projections resulting from differences among IAMs and GCMs on each scenario (Popp et al., 2017). The gaps identified through BES-SIM and future directions for research and modelling will be published separately as well as analyses of the results on the model intercomparison and on individual models.

As a long-term perspective, BES-SIM is expected to provide critical foundation and insights for the ongoing development of nature-centred, multiscale Nature Futures scenarios (Rosa et al., 2017). Catalysed by the IPBES Expert Group on Scenarios and Models, this new scenarios and modelling framework will shift traditional ways of forecasting impacts of society on nature to more integrative, biodiversity-centred visions and pathways of socio-economic and ecological systems. A future round of BES-SIM could use these biodiversity-centred storylines to project dynamics of biodiversity and ecosystem services and associated consequences for socio-economic development and human well-being. This will help policymakers and practitioners to collectively identify pathways for sustainable futures based on alternative biodiversity management approaches and assist researchers in incorporating the role of biodiversity in socio-economic scenarios.

**9. Code and data availability**

The output data from this model intercomparison will be downloadable from the website of the IPBES Expert Group on Scenarios and Models in the future (https://www.ipbes.net/deliverables/3c-scenarios-and-modelling). The LUH2 land-use data used for model runs are available on http://luh.umd.edu/data.shtml. The climate datasets used in BES-SIM can be downloaded from the respective websites (https://www.isimip.org/outputdata/, http://worldclim.org/version1)

There is a Supplement for this manuscript with Supplementary Methods and Tables S1, S2, S3, which can be found here: https://www.geosci-model-dev-discuss.net/gmd-2018-115/.

*Author contributions. All authors co-designed the study under the coordination of Henrique M. Pereira, Rob Alkemade, Paul Leadley and Isabel M.D. Rosa. HyeJin Kim prepared the manuscript with contributions from all co-authors.*

*Competing interests: The authors declare that they have no conflict of interest.*

*Acknowledgements*

HJK, ISM, FW, CG and HMP are supported by the German Centre for integrative Biodiversity Research (iDiv) Halle-Jena-Leipzig, funded by the German Research Foundation (FZT 118). IMDR acknowledges funding from the European Union's Horizon 2020 research and innovation programme under the Marie Sklodowska-Curie grant agreement No 703862. PL is supported by the LabEx BASC supported by the French "Investment d'Avenir" program (grant ANR-11-LABX-0034). GCH and LPC gratefully acknowledge the support of DOE-SciDAC program (DE‑SC0012972). AA, AK, BQ and PA acknowledge support from the Helmholtz Association and its ATMO Programme, and EU FP7 project LUC4C. AP, ADP and SLLH are supported by the Natural Environment Research Council U.K. (grant number NE/M014533/1) and by a DIF grant from the Natural History Museum. RCK and RS are supported by private gifts to the Natural Capital Project. DL, FDF, PH, and MO are supported by the project IS-WEL-Integrated Solutions for Water, Energy and Land funding from Global Environmental Facility, Washington, USA, coordinated by United Nations Industrial Development Organization (UNIDO), UNIDO Project No. 140312. FDF and MO are supported by the ERC SYNERGY grant project IMBALANCE-P-Managing Phosphorous limitation in a nitrogen-saturated Anthropocene, funding from European Commission, European Research Council Executive Agency, grant agreement No. 610028. DL and PH are supported by the project SIGMA- Stimulating Innovation for Global Monitoring of Agriculture and its Impact on the Environment in support of GEOGLAM, funding from the European Union's FP7 research and innovation programme under the Environment area, grant agreement No. 603719. TH, HO, AH, SF, TM and KT are supported by the Global Environmental Research (S-14) of the Ministry of the Environment of Japan. TH, SF and KT are supported by Environment Research and Technology Development Fund 2-1702 of the Environmental Restoration and Conservation Agency of Japan. MH is supported by a KR Rasmussen Foundation grant "Modelling the Biodiversity Planetary Boundary and Embedding Results into Policy" (FP-1503-01714). VH acknowledges support from the Earth Systems and Climate Change Hub, funded by the Australian Government's National Environmental Science Program. CM acknowledges funding from NSF Grant DEB1565046. Finally, we also thank the following organizations for funding the workshops: PBL Netherland Environment Assessment Agency, UNESCO (March 2016), iDiv German Centre for Integrative Biodiversity Research (October 2016, October 2017) and Zoological Society of London (January 2018).

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

**Figure 1: Input-models-output flowchart of BES-SIM.**

| Input -<br>Harmonized scenarios & data<br>*(sections 2, 3)* | Models<br>*(section 4)* | Output -<br>Common (or categorized) metrics<br>*(section 5)* |
|---|---|---|
| Land Use<br>LUH2<br>(native resolution or GLOBIO downscaled) for<br>SSP1, SSP3, SSP5<br><br>Climate<br>ISIMIP2a IPSL-CM5A-LR<br>(native resolution or WorldClim downscaled) or MAGICC* for<br>RCP2.6, RCP6.0, RCP8.5<br>*GLOBIO<br><br>Others (model-specific)<br>Species records, habitat affinities, range maps, vegetation cover, population density, correlation coefficients of pressure drivers, etc.<br><br>*(see Tables 2, 3, 5 and S1)* | Biodiversity<br>*Species-based:*<br>AIM-biodiversity, InSiGHTS, MOL, BIOMOD2 (uncertainty analysis)<br><br>*Community-based:*<br>cSAR-iDiv, cSAR-IIASA-ETH, BILBI, PREDICTS,<br>GLOBIO - Aquatic, Terrestrial<br><br>*Ecosystem-based:*<br>Madingley<br><br><br>Ecosystem Functions and Services<br>LPJ-GUESS, LPJ, CABLE (DGVMs), GLOBIO-ES, InVEST, GLOSP<br><br>*(see Appendix 1 and Table S2)* | Biodiversity:<br>Local, regional and global diversity<br>Abundance and intactness<br>Local and global habitat change<br><br><br>Nature's Contributions to People:<br>Pollination<br>Climate regulation<br>Water regulation (quantity, quality)<br>Soil protection<br>Hazards/extreme events regulation<br>Pest control<br>Energy production<br>Food and feed<br>Materials<br><br>*(see Tables 4, 6, and S3)* |

**Table 1: Characteristics of (a) SSP and (b) RCP scenarios simulated in BES-SIM** (adapted from Moss et al., 2010; O'Neill et al., 2017; Popp et al., 2017; van Vuuren et al., 2011).

(a) SSP scenarios

| | SSP1 Sustainability | SSP3 Regional Rivalry | SSP5 Fossil-fueled Development |
|---|---|---|---|
| Population growth | Relatively low | Low (OECD countries) to high (high fertility countries) | Relatively low |
| Urbanization | High | Low | High |
| Equity and social cohesion | High | Low | High |
| Economic growth | High to medium | Slow | High |
| International trade and globalization | Moderate | Strongly constrained | High |
| Land-use regulation | Strong to avoid environmental trade-off | Limited with continued deforestation | Medium with slow decline in deforestation |
| Agricultural productivity | High improvements with diffusion of best practices | Low with slow technology development and restricted trade | Highly managed and resource intensive |
| Consumption & diet | Low growth in consumption, low-meat | Resource-intensive consumption | Material-intensive consumption, meat-rich diet |
| Environment | Improving | Serious degradation | Highly successful management |
| Carbon intensity | Low | High | High |
| Energy intensity | Low | High | High |
| Technology development | Rapid | Slow | Rapid |
| Policy focus | Sustainable development | Security | Development, free market, human capital |
| Participation of the land-use sector in mitigation policies | Full | Limited | Full |
| International cooperation for climate change mitigation | No delay | Heavy delay | Delay |
| Institution effectiveness | Effective | Weak | Increasingly effective |

(b) RCP scenarios

| | RCP2.6 Low emissions | RCP6.0 Intermediate emissions | RCP8.5 High emissions |
|---|---|---|---|
| Radiative forcing | Peak at $3W/m^2$ before 2100 and decline | Stabilizes without overshoot pathways to $6W/m^2$ in 2100 | Rising forcing pathways leading to $8.5 W/m^2$ in 2100 |
| Concentration (p.p.m) | Peak at 490 $CO_2$ equiv. before 2100 and then declines | 850 $CO_2$ equiv. (at stabilization after 2100) | >1,370 $CO_2$ equiv. in 2100 |
| Methane emission | Reduced | Stable | Rapid increase |
| Reliance on fossil fuels | Decline | Heavy | Heavy |
| Energy intensity | Low | Intermediate | High |
| Climate policies | Stringent | Very modest to almost none | High range of no policies |

(c) SSPxRCP scenarios

| | SSP1xRCP2.6 Highest mitigation | SSP3xRCP6.0 Limited mitigation | SSP5xRCP8.5 No mitigation |
|---|---|---|---|
| Bioenergy | Low | Highest | Lowest |

**Table 2: Sources of land use and climate input data in BES-SIM.**

| BES-SIM model | Land-use data | | Climate data | | |
|---|---|---|---|---|---|
| | LUH2 v2.0 Native resolution *0.25 degree* | LUH2 v2.0 Downscaled (GLOBIO) *300m* | ISIMIP2a IPSL-CM5A-LR Native resolution *0.5 degree* | ISIMIP2a IPSL-CM5A-LR Downscaled (WorldClim) *1km* | IMAGE† (MAGICC 6.0) |
| **Species-based models of biodiversity** | | | | | |
| AIM-biodiversity | * | | * | | |
| InSiGHTS | * | | | * | |
| MOL | * | | | * | |
| **Community-based models of biodiversity** | | | | | |
| cSAR-iDiv | * | | | | |
| cSAR-IIASA-ETH | * | | | | |
| BILBI | * | | | * | |
| PREDICTS | * | | | | |
| GLOBIO - Aquatic | * | | | | * |
| GLOBIO4 - Terrestrial | | * | | | * |
| **Ecosystems-based model of biodiversity** | | | | | |
| Madingley | * | | * | | |
| **Models of ecosystem functions and services** | | | | | |
| LPJ-GUESS | * | | * | | |
| LPJ | * | | * | | |
| CABLE | * | | * | | |
| GLOBIO-ES | * | | | | * |
| InVEST | | * | | * | |
| GLOSP | * | | * | | |

†All GLOBIO models use MAGICC climate data from the IMAGE model.

**Table 3: Improvements made in the Land Use Harmonization v2 (LUH2) from LUH v1** (sources: Hurtt et al., 2011; Hurtt et al., 2018).

| | LUH v1 | LUH v2 |
|---|---|---|
| Spatial resolution | 0.5 degree | 0.25 degree |
| Time steps | Annually from 1500 to 2100 | Annually from 850 to 2100 |
| Land use categories | 5 categories<br>Primary<br>Secondary<br>Pasture<br>Urban<br>Crop | 12 categories<br>Forested primary land (primf)<br>Non-forested primary land (primn)<br>Potentially forested secondary land (secdf)<br>Potentially non-forested secondary land (secdn)<br>Managed pasture (pastr)<br>Rangeland (range)<br>Urban land (urban)<br>C3 annual crops (c3ann)<br>C3 perennial crops (c3per)<br>C4 annual crops (c4ann)<br>C4 perennial crops (c4per)<br>C3 nitrogen-fixing crops (c3nfx) |
| Future | RCPs (4)<br>2.6<br>4.5<br>6.0<br>8.5 | SSPs (6)<br>SSP1-RCP2.6<br>SSP4-RCP3.4<br>SSP2-RCP4.5<br>SSP4-RCP6.0<br>SSP3-RCP7.0<br>SSP5-RCP8.5 |
| Land use transitions | <20 per grid cell per year | >100 per grid cell per year |
| Improvements | | - New shifting cultivation algorithm<br>- Landsat forest/non-forest change constraint<br>- Expanded diagnostic package<br>- New historical wood harvest reconstruction<br>- Agricultural management layers: irrigation, fertilizer, biofuel crops, wood harvest product split, crop rotations, flooded (rice) |

**Table 4: Selected output indicators for intercomparison of biodiversity and ecosystems models.** For species diversity change, both proportional changes in species richness (P) and absolute changes (N) are reported. Some models project the α metrics at the level of the grid cell (e.g. species-based and SAR based community models) while others average the local values of the metrics across the grid cell weighted by the area of the different habitats in the cell (e.g. PREDICTS, GLOBIO).

| BES-SIM model | Species diversity change at local scale ($P\alpha$ and $N\alpha$) | Species diversity change at subregional and global scale ($P\gamma$ and $N\gamma$) | Abundance-based intactness at local scale ($I\alpha$) | Mean habitat extent change at local and global scale ($H\alpha$ and $H\gamma$) |
|---|---|---|---|---|
| **Species-based models of biodiversity** | | | | |
| AIM-biodiversity | * | * | | * |
| InSiGHTS | * | * | | * |
| MOL | * | * | | * |
| **Community-based models of biodiversity** | | | | |
| cSAR-iDiv | * | * | | |
| cSAR-IIASA-ETH | * | * | | |
| BILBI | | * | | |
| PREDICTS | * | | * | |
| GLOBIO - Aquatic | | | * | |
| GLOBIO - Terrestrial | | | * | |
| **Ecosystems-based model of biodiversity** | | | | |
| Madingley | | | * | |

**Table 5: Scenario (forcing data) for models in BES-SIM.**

| BES-SIM model | Historical | Future Land-Use Change or Climate (2050) | | |
|---|---|---|---|---|
| | | Land use only, climate held constant at 2015 (SSP1, SSP3, SSP5) | Climate change only, land use held constant at 2015 (RCP2.6, RCP6.0, RCP8.5) | Land use and climate (SSP1xRCP2.6, SSP3xRCP6.0, SSP5xRCP8.5) |
| **Species-based models of biodiversity** | | | | |
| AIM-biodiversity | * | * | * | * |
| InSiGHTS | * | * | * | * |
| MOL | | * | * | * |
| **Community-based models of biodiversity** | | | | |
| cSAR-iDiv | * | * | | |
| cSAR-IIASA-ETH | * | * | | |
| BILBI | * | * | | * |
| PREDICTS | * | * | | |
| GLOBIO - Aquatic | | | | * |
| GLOBIO - Terrestrial | | * | * | * |
| **Ecosystems-based model of biodiversity** | | | | |
| Madingley | * | | | * |
| **Models of ecosystem functions and services** | | | | |
| LPJ-GUESS | * | * | * | * |
| LPJ | * | * | * | * |
| CABLE | * | * | * | * |
| GLOBIO-ES | * | | | * |
| InVEST | * | | | * |
| GLOSP | | | | * |

**Table 6: Selected output indicators for inter-comparison of ecosystem functions and services models, categorized based on the classification of Nature's Contributions to People (Díaz et al., 2018).**

| BES-SIM model | NCP 2. Pollination and dispersal of seeds and other propagules | NCP 4. Regulation of climate | NCP 6. Regulation of freshwater quantity, location and timing | NCP 7. Regulation of freshwater and coastal water quality | NCP 8. Formation, protection and decontamination of soils and sediments | NCP 9. Regulation of hazards and extreme events | NCP 10. Regulation of detrimental organisms and biological processes | NCP 11. Energy | NCP 12. Food and feed | NCP 13. Materials, companionship and labor |
|---|---|---|---|---|---|---|---|---|---|---|
| LPJ-GUESS | | Total carbon Vegetation carbon | Monthly runoff | Nitrogen leaching | | | | Bioenergy-crop production | Harvested carbon in croplands that are used for food production | Wood harvest (LUH2 extraction) |
| LPJ | | Total carbon Vegetation carbon | Monthly runoff | | | | | | | |
| CABLE | | Total carbon Vegetation carbon | Monthly runoff, Total runoff | | | | | | Above ground carbon removed from cropland and pastures as a result of harvest and grazing | Wood harvest |
| GLOBIO-ES | Fraction of cropland potentially pollinated, relative to all available cropland | Total carbon | Water scarcity index | Nitrogen in water Phosphorus in water | Erosion protection: fraction with low risk relative to the area that needs protection | Flood risk: number of people exposed to river flood risk | Pest control: Fraction of cropland potentially protected, relative to all available cropland | | Total crop production Total grass production | |
| InVEST | Proportion of agricultural lands whose pollination needs are met | | | Nitrogen export Nitrogen export*capita | | Coastal vulnerability Coastal vulnerability *capita | | | Caloric production per hectare on the current landscape for each crop type | |
| GLOSP | | | | | Soil protection | | | | | |

# Appendix 1

**Table A1: Description of biodiversity and ecosystem functions and services models in BES-SIM.**

| BES-SIM Model | Brief model description | Defining features and key processes | Model modification | Spatial resolution | Time steps | Taxonomic or functional scope | Key reference |
|---|---|---|---|---|---|---|---|
| **Species-based models of biodiversity** | | | | | | | |
| AIM-biodiversity (Asia-Pacific Integrated Model – biodiversity) | A species distribution model that estimates biodiversity loss based projected shift of species range under the conditions of land use and climate change. | Distribution of suitable habitat (land) estimated from climate and land-use data using a statistical model on species presence and climate and land-use classifications, calibrated by historical data. | Please see Table S2 for detailed methodology. | 0.5 degree | 1900, 2015, 2050, 2070 | Amphibians, birds, mammals, plants, reptiles | (Ohashi et al., submitted) |
| InSiGHTS | A high-resolution, cell-wise, species-specific hierarchical species distribution model that estimate the extent of suitable habitat (ESH) for mammals accounting for land and climate suitability. | Bioclimatic envelope models fitted based on ecologically current reference bioclimatic variables. Species' presence and pseudo-absence records from sampling within and outside of species' ranges. Forecasted layers of land-use/land-cover reclassified according to expert-based species-specific suitability indexes. | Increased number of modelled species, new scenarios for climate and land use. | 0.25 degree | 1900, 2015, 2050, 2070 | Mammals | (Rondinini et al., 2011; Visconti et al., 2016) |
| MOL (Map of Life) | An expert map based species distribution model that projects potential losses in species occurrences and geographic range sizes given changes in suitable conditions of climate and land cover change. | Expert maps for terrestrial amphibians, birds and mammals as baseline for projections, combined with downscaled layers for current climate. A penalized point process model estimated individual species niche boundaries, which were projected into 2050 and 2070 to estimate range loss. Species habitat preference-informed land cover associations were used to refine the proportion of suitable habitat in climatically suitable cells with present and future land-cover based projections. | Inductive species distribution modelling was built using point process models to delineate niche boundaries. Binary maps of climatically suitable cells were rescaled (to [0,1]) based on the proportion of the cell within a species land cover preference | 0.25 degree | 2015, 2050, 2070 | Amphibians, birds, mammals | (Jetz et al., 2007; Merow et al., 2013) |

| BES-SIM Model | Brief model description | Defining features and key processes | Model modification | Spatial resolution | Time steps | Taxonomic or functional scope | Key reference |
|---|---|---|---|---|---|---|---|
| BIOMOD2 (BIOdiversity MODelling) | An R-package that allows running up to nine different algorithms of species distribution models using the same data and the same framework. An ensemble could then be produced allowing a full treatment of uncertainties given the data, algorithms, climate models, climate scenarios. | BIOMOD2 is based on species distribution models that link observed or known presence-absence data to environmental variables (e.g. climate). Each model is cross-validated several times (a random subset of 70% of the data is used for model calibration while 30% are hold out for model evaluation). Models are evaluated using various metrics. | | 100km | 2015, 2050, 2070 | Amphibians, birds, mammals | (Thuiller, 2004; Thuiller et al., 2009, 2011) |

| Community-based models of biodiversity | | | | | | | |
|---|---|---|---|---|---|---|---|
| cSAR (Countryside Species Area Relationship) - iDiv | A countryside species-area relationship model that estimates the number of species persisting in a human-modified landscape, accounting for the habitat preferences of different species groups. | Proportional species richness of each species group is a power function of the sum of the areas of each habitat in a landscape, weighted by the affinity of each species group to each habitat type. Species richness is calculated by multiplying the proportional species richness by the number of species known to occur in the area. Total number of species in a landscape is the sum of the number of species for each species group. | Two functional groups of bird species: (1) forest birds; (2) non-forest birds. Habitat affinities retrieved from PREDICTS database. | 0.25 degree | 1900-2010 (10 years interval), 2015, 2050, 2070, 2090 | Birds (forest, non-forest, all) | (Martins and Pereira, 2017) |
| cSAR-IIASA-ETH | A countryside species area relationship model that estimates the impact of time series of spatially explicit land-use and land-cover changes on community-level measures of terrestrial biodiversity. | Extends concept the SAR to mainland environment where the habitat size depends not only on the extent of the original pristine habitat, but also on the extent and taxon-specific affinity of the other non-pristine land uses and land covers (LULC) of conversion. Affinities derived from field records. Produces the average habitat suitability, regional species richness, and loss of threatened and endemic species for five taxonomic groups. | Refined link between LULCC and habitat (gross transitions between LULC classes at each time) and better accounting of time dynamics of converted LULC classes. | 0.25 degree | 1500-1900 (100 years interval), 1900-2090 (10 years interval) | Amphibians, birds, mammals, plants, reptiles | (Chaudhary et al., 2015; UNEP, 2016) |

| BILBI (Biogeographic modelling Infrastructure for Large-scale Biodiversity Indicators) | A modelling framework that couples application of the species-area relationship with correlative generalized dissimilarity modeling (GDM)-based modelling of continuous patterns of spatial and temporal turnover in the species composition of communities (applied in this study to vascular plant species globally). | The potential effects of climate scenarios on beta-diversity patterns are estimated through space-for-time projection of compositional-turnover models fitted to present-day biological and environmental data. These projections are then combined with downscaled land-use scenarios to estimate the proportion of species expected to persist within any given region. This employs an extension of species-area modelling designed to work with biologically-scaled environments varying continuously across space and time. | Please see Table S3 for detailed methodology. | 1 km (30 arcsec) | 1900, 2015, 2050 | Vascular plants | (Ferrier et al., 2004, 2007) |
| PREDICTS (Projecting Responses of Ecological Diversity In Changing Terrestrial Systems) | The hierarchical mixed-effects model that estimates how four measures of site-level terrestrial biodiversity – overall abundance, within-sample species richness, abundance-based compositional similarity and richness-based compositional similarity – respond to land use and related pressures. | Models employ data from the PREDICTS database encompassing 767 studies from over 32,000 sites on over 51,000 species. Models assess how alpha diversity is affected by land use, land-use intensity and human population density. Model coefficients are combined with past, present and future maps of the pressure data to make global projections of response variables, which are combined to yield the variants of the Biodiversity Intactness Index (an indicator first proposed by (Scholes and Biggs, 2005)). | PREDICTS LU classes recurated for LUH2. Abundance rescaled within each study. Baseline of minimally-used primary vegetation. Compositional similarity models included human population. Study-level mean human population and agricultural suitability used as control variables. Proximity to road omitted. | 0.25 degree | 900-2100 | All | (Newbold et al., 2016; Purvis et al., 2018) |

| | | | | | | | |
|---|---|---|---|---|---|---|---|
| GLOBIO (GLObal BIOdiversity) - Aquatic | A modelling framework that quantifies the impacts of land-use, eutrophication, climate change and hydrological disturbance on freshwater biodiversity, quantified as the mean species abundance (MSA) and ecosystem functions/services. | Comprises a set of (mostly correlative) relationships between anthropogenic drivers and biodiversity/ES of rivers, lakes and wetlands. Based on the catchment approach, i.e., the pressures on the aquatic ecosystems are based on what happens in their catchment. Based on the literature. | | 0.5 degree | 2015, 2050 | All | (Janse et al., 2015, 2016) |
| GLOBIO - Terrestrial | A modelling framework that quantifies the impacts of multiple anthropogenic pressures on local biodiversity (MSA). | Based on a set of correlative relationships between biodiversity (MSA) on the one hand and anthropogenic pressures on the other, quantified based on meta-analyses of biodiversity data reported in the literature. Georeferenced layers of the pressure variables are then combined with the response relationships to quantify changes in biodiversity. | Improved land-use allocation routine, improved response relationships for encroachment (hunting) | 10 arc-seconds (~300 m) | 2015, 2050 | All | (Schipper et al., 2016) |

| **Ecosystems-based model of biodiversity** | | | | | | | |
|---|---|---|---|---|---|---|---|
| Madingley | An integrated process-based, mechanistic, general ecosystem model that uses a unified set of fundamental ecological concepts and processes to predict the structure and function of the ecosystems at various levels of organisation for marine or terrestrial. | Grouped by heterotroph cohorts, organisms are defined by functional traits rather than the taxonomy. Heterotrophs, defined by categorical (trophic group; hermoregulation strategy; reproductive strategy) and quantitative (current body mass; mass at birth; and mass at reproductive maturity) traits are modelled as individuals dynamically. Simulates the autotroph ecological processes of growth and mortality; and heterotroph metabolism, eating, reproduction, growth, mortality, and dispersal. Dispersal is determined by the body mass. | Incorporation of temporally changing climate, and natural and human impacted plant stocks to better represent the LUHv2 land-use projections. Calculation of functional diversity and dissimilarity to represent community changes | 1 degree | 1901, 1915-2070 (5 years interval) | Three functional groups | (Harfoot et al., 2014b) |
| **Models of ecosystem functions and services** | | | | | | | |
| LPJ-GUESS (Lund-Potsdam-Jena General Ecosystem Simulator) | A process-based "demography enabled" dynamic global vegetation model that computes vegetation and soil state and function, as well as distribution of vegetation units dynamically in space and time in response to climate change, land-use change and N-input. | Vegetation dynamics result from growth and competition for light, space and soil resources among woody plant individuals and herbaceous understorey. A suite of simulated patches per grid cell represents stochastic processes of growth and mortality (succession). Individuals for woody plant functional types (PFTs) are identical within an age-cohort. Processes such as photosynthesis, respiration, stomatal conductance are simulated daily. Net primary production (NPP) accrued at the end of each simulation year is allocated to leaves, fine roots and, for woody PFTs, sapwood, resulting in height, diameter and biomass growth. | The model version used here has some updates to the fire model compared to Knorr et al. (2016) see also Rabin et al. (2017). Simulations also accounted for wood harvest, using the modelled recommendations from LUH2. | 0.5 degree | 1920, 1950, 1970, 2015, 2050, 2070 | | (Lindeskog et al., 2013; Olin et al., 2015; Smith et al., 2014) |

| | | | | | | | |
|---|---|---|---|---|---|---|---|
| LPJ (Lund-Potsdam-Jena) | A big leaf model that simulates the coupled dynamics of biogeography, biogeochemistry and hydrology under varying climate, atmospheric $CO_2$ concentrations, and land-use land cover change practices to represent demography of grasses and trees in a scale from individuals to landscapes. | Hierarchical representation of the land surface - tiles represent land use with various plant or crop functional types. Implements establishment, mortality, fire, carbon allocation, and land cover change on annual time steps, and calculates photosynthesis, autotrophic respiration, and heterotrophic respiration on daily time steps. Fully prognostic, meaning that PFT distributions and phenology are simulated based on physical principles within a numerical framework. | LPJ represents the full set of states and transitions represented in LUHv2 and improved estimate of carbon fluxes from land-cover change. | 0.5 degree | 1920, 1950, 1970, 2015, 2050, 2070 | | (Poulter et al., 2011; Sitch et al., 2003) |
| CABLE (Community Atmosphere Biosphere Land Exchange) | A "demography enabled" global terrestrial biosphere model that computes vegetation and soil state and function dynamically in space and time in response to climate change, land-use change and N-input. | Combines biophysics (coupled photosynthesis, stomatal conductance, canopy energy balance) with daily biogeochemical cycling of carbon and nitrogen (CASA-CNP) and annual patch-based representation of vegetation structural dynamics (POP). Accounts for gross land-use transitions and wood harvest, including effects on patch age distribution in secondary forest. Simulates co-ordination of rate-limiting processes in C3 photosyntheisis, as an outcome of fitness maximisation. | | 1 degree | 1920, 1950, 1970, 2015, 2050, 2070 | | (Haverd et al., 2017) |

| GLOBIO-Ecosystem Services | The model simulates the influence of various anthropogenic drivers on ecosystem functions and services. | Quantifies a range of provisioning services (e.g. crop production, grass and fodder production, wild food), regulating services (e.g. pest control, pollination, erosion risk reduction, carbon sequestration), and culture services (e.g. nature based tourism) and other measures (e.g. water availability, food risk reduction, harmful algal blooms). Derived from various models, including the Integrated Model to Assess the Global Environment (IMAGE) model and PCRaster Global Water Balance (PCR-GLOBWB), and from empirical studies using meta-analysis. | Relationships between land use and the presence of pollinators and predators updated through additional peer review papers. | 0.5 degree | 2015, 2050, 2070 | | (Alkemade et al., 2009, 2014; Schulp et al., 2012) |
|---|---|---|---|---|---|---|---|
| InVEST (Integrated Valuation of Ecosystem Services and Tradeoffs) | A suite of geographic information system (GIS) based spatially-explicit models used to map and value the ecosystem goods and services in biophysical or economic terms. | 18 models for distinct ecosystem services designed for terrestrial, freshwater, marine and coastal ecosystems. Based on production functions that define how changes in an ecosystem's structure and function are likely to affect the flows and values of ecosystem services across a land- or a seascape. Accounts for both service supply and the location and activities of demand. Modular and selectable. | The crop-production model was simplified from 175 crops to the 5 crop-types reported in LUH2. Other models have minor simplifications; see tables S2 and S3 for more detail. | 300m and 5 arc-minute | 2015, 2050 | | (Arkema et al., 2013; Chaplin-Kramer et al., 2014; Guannel et al., 2016; Johnson et al., 2014, 2016; Redhead et al., 2018; Sharp et al., 2016) |

| GLOSP (GLObal Soil Protection) | A 2D soil erosion model based on the Universal Soil Loss Equation that uses climate and land-use projections to estimate global and local soil protection. | Protected soil (Ps) is defined as the amount of soil that is prevented from being eroded (water erosion) by the mitigating effect of available vegetation. Ps is calculated from the difference between soil erosion (Se) and potential soil erosion (Pse) based on the integration of the joint effect of slope length, rainfall erosivity, and soil erodibility. Soil protection is given by the value of fractional vegetation cover calculated as a function of land use, altitude, precipitation, and soil properties. | Please see Table S3 for detailed methodology. | 0.25 degree | 2015, 2050 | | (Guerra et al., 2016) |
|---|---|---|---|---|---|---|---|

**Appendix 2**

**List of Acronyms**

| | |
|---|---|
| AIM | Asia-pacific Integrated Model |
| BES-SIM | Biodiversity and Ecosystem Services Scenario-based Intercomparison of Models |
| BIOMOD | BIOdiversity MODelling |
| BILBI | Biogeographic modelling Infrastructure for Large-scale Biodiversity Indicators |
| CABLE | Community Atmosphere Biosphere Land Exchange |
| CMIP | Climate Model Inter-comparison Project |
| cSAR | Countryside Species Area Relationship |
| DGVM | Dynamic Global Vegetation Model |
| EBV | Essential Biodiversity Variable |
| ESM | Earth System Models |
| GBIF | Global Biodiversity Information Facility |
| GBO | Global Biodiversity Outlooks |
| GCM | General Circulation Models |
| GEO | Global Environmental Outlook |
| GLOBIO | GLObal BIOdiversity |
| GLOSP | GLObal Soil Protection |
| IAM | Integrated Assessment Models |
| IMAGE | Integrated Model to Assess the Global Environment |
| InVEST | Integrated Valuation of Ecosystem Services and Tradeoffs |
| IPBES | Intergovernmental Science-Policy Platform on Biodiversity and Ecosystem Services |
| IPCC | Intergovernmental Panel on Climate Change |
| IPSL-CM5A-LR | Institut Pierre-Simon Laplace-Climate Model 5A-Low Resolution |
| ISIMIP | Inter-Sectoral Impact Model Intercomparison Project |
| LPJ | Lund-Potsdam-Jena |
| LPJ-GUESS | Lund-Potsdam-Jena General Ecosystem Simulator |
| LUH2 | Land Use Harmonization Project version 2 |
| MA | Millennium Ecosystem Assessment |
| MAgPIE | The Model of Agricultural Production and its Impact on the Environment |
| MIP | Model Intercomparison Project |
| MOL | Map of Life |
| NCP | Nature's Contributions to People |
| REMIND | Regionalized Model of Investments and Development |
| PREDICTS | Projecting Responses of Ecological Diversity In Changing Terrestrial Systems |
| RCM | Regional Climate Models |
| RCPs | Representative Concentration Pathways |
| PCR-GLOBWB | PCRaster Global Water Balance |
| SAR | Species Area Relationship |
| SR | Species Richness |
| SSPs | Shared Socio-economic Pathways |

