# Peer review of "A protocol for an intercomparison of biodiversity and ecosystem services models using harmonized land-use and climate scenarios"

_Geoscientific Model Development, 2018_

## Referee Comment (RC1) · Anonymous Referee #1 · 30 Aug 2018

The manuscript outlines a protocol for comparing models of the impacts of land-use and climate on biodiversity and ecosystem services. Intercomparisons of biodiversity and ecosystem services models and scenarios is much needed, especially for inter-governmental processes such as the IPBES, other policy processes, and conservation interventions. The manuscript outlines a process of comparing 16 spatially-explicit models for past (up to 900 AD), present (2015) and future (up to 2070) based on 3 scenarios (combinations of SSPs and RCPs) and the output variables that can be compared. Overall, biodiversity and ecosystem services model intercomparisons are much needed and this manuscript outlines a protocol for such intercomparisons for the first time.

Uncertainty is a critical part of models and model intercomparisons as acknowledged in the manuscript (section 7). The section on uncertainties, how uncertainty will be assessed within models and across models is too brief to be helpful. It would be interesting to better understand what the "comprehensive uncertainty analysis based on a variance partitioning approach" would involve. Furthermore, the text states that uncertainty of "the models of biodiversity" (P11) will be assessed, but there is no mention on how ecosystem services model outputs will be assessed for uncertainty. Both types of models will contain uncertainties that require assessment.

The section on other input data (section 3.3) should acknowledge the need for additional parameters within each model, in particular in ecosystem service models. For example, InVEST requires detailed information on parameters/look up tables to allocate the ecosystem service; Madingley has predator-prey relationships encoded. Will the default values be used for the intercomparisons or will models be modified? Which versions of the models will be used? Some of this information is provided in Appendix 1, however more detail could be provided.

Minor comments

P2L27: ecosystems are a subset of biodiversity as defined by the Convention on Biological Diversity, therefore delete "ecosystems" here.

P2L31-32: the statement that land-use change has immediate impacts on biodiversity and ecosystem services and impacts of climate change involves time lags is not correct. Both land use and climate changes can have immediate and lagged impacts. There is substantial evidence that climate change can have immediate impacts (e.g. Wellbergen et al. 2008 Proc Roy Soc B 275: 419-425) and land use impacts can be time lagged (e.g. McMichael et al. 2017. Ancient human disturbances may be skewing our understanding of Amazonian forests. PNAS 114: 522-527; Jakovac et al. 2016. Land use as a filter for species composition in Amazonian secondary forests. J. Veg. Sci. 27: 1104-1116; Graham et al. Graham et al. 2017. Implications of afforestation
for bird communities: the importance of preceding land-use type. Biodiv. Cons. 26: 3051-3071).

P3L8 Ferrier et al. 2016 missing from reference list, maybe should be IPBES 2016?

P5L21 Need to clarify the difference between RCP7.0 which was used for land-use projections and RCP6.0 which is used for other scenario production. Explain why SSP3/RCP7.0 was not used instead of the mixed SSP3/RCP6.0+RCP7.0 for land use. Furthermore, Table 5 does not show the use of RCP7.0 for land use, it is shown as RCP6.0; check this is correct.

P6L16: spell out ESM at first use.

P7L21: Table 2 shows 13 (and not 12) models requiring climate data. Which is correct, text or table?

P10L12-15: reword this sentence, not comprehensible.

P10L30: "units of the metrics" are not listed in Table 6. Update Table 6 with units, or reword text.

P11L3: replace "Additional" with "Additionally"

P11L4: Table 5 does not show the multiple time points from past to future, this information is provided in Appendix 1 first table. Note tables in Appendix 1 do not have legends or numbers.

P12L4-5: insert "to" before "the CBD and. . ." and before "other relevant stakeholders"

P14- References: some references included "edited by" information for journal articles, e.g. Harfoot et al 2014b, Heinimann et al 2017. Check this is in line with the journal reference guidelines. Several references are submitted or in preparation, hence make it impossible to fully assess this manuscript.

P23-all tables: check carefully throughout. $CO_2$ and $m^2$ should have subscript and

superscript "2"s.

P23Table1: Information on RCP6.0 is provided, however as RCP7.0 is used for land use projection, some info needs to be provided for RCP7.0 in this table or elsewhere. Information on climate policies is missing for RCP6.0. Table legend should read "Sources of land use and climate input data in BES-SIM" as other input data are used in all models (see later Tables).

P26Table4: better explain "alpha and gamma metrics".

P32: Scholes et al. 2005 reference is missing in reference list

P34: spell out PFTs at first use (and all other acronyms throughout, e.g. GIS on P36, etc.)

P37: PCR-GLOBWB is missing from the list of acronyms

P1-37: throughout the text reference is being made to Table S1, S2 etc. (e.g. P7L11), however no tables S1, S2 etc. are included.

---

## Referee Comment (RC2) · Anonymous Referee #2 · 31 Aug 2018

Dear authors

The manuscript "A protocol for an intercomparison of biodiversity and ecosystem services models using harmonized land-use and climate scenarios" could be suitable for Geoscientific Model Development. This manuscript explains the framework of BES-SIM (inter-comparison study). This model inter-comparison study tries to contribute not only to science communities including earth system modeling, climate science, and ecology, but also the other stakeholders such as policy maker. I fully agreed on the importance of this project to manage the impact on biodiversity and ecosystem

services by anthropogenic activities. However, I feel the main manuscript is the lack of content to follow the overall picture of BES-SIM.

From the view to GMD paper, I cannot recommend acceptance for the publication in the current manuscript.

Major comments

Many concerns are as follow;

1. For outside of the ecological area of expertise, the manuscript is not so helpful to understand BES-SIM. For example, there are many outputs related to the biodiversity, however, there are few definition and explanation of each output. At least, this journal's fields is not ecology. So, readers cannot follow even the unit of outputs provided in BES-SIM (e.g., abundance). They should be clearly described in the main manuscript.

2. (I guess) To each SSP (RCP), the narratives (and interpretation) from the view to biodiversity and its social governance should be described in the scenario session. Table 1 provides such kind of information against each factor. The readers want to know these narratives in advance. For example, which one is business as usual to biodiversity? If there are no narratives in biodiversity (it means biodiversity sector is just passive against the other policies), please explain instead. Or, please make the narratives so as to follow the each SSP concept.

   (Sorry, I wrote this comment before reading the discussion. But, even after the reading discussion, I keep this comment. Also, I apologize that I cannot read Rosa et al. (2017) due to the non-license to read.)

**[GMDD](...)**

Interactive
comment

Individual comments

**P2L17–19** Please suggest the combination (i.e., SSP1×RCP2.6?) here.

**P3L2** Please clarify "at high levels of climate change".

**P3L5–6** Please suggest how to assess the impacts using the scenario? With empirical or mechanistic model?

**P3L6–8 Models are ...** Meaning of this sentence is not clear.

**P3L9** I'm not sure "cross-model harmonization".

**P3L9–12** "addressing this issue" <- Please add the citation.

**P3L21–23** I didn't find Ferrier et al. (2016) said the meaning of this sentence in the discussion. Is this a correct citation?

**P3L25–27** I cannot agree with the sentence "to improve the robustness and comprehensiveness". MIPs (at least without observation data) will not contribute to the improvement of the robustness. Did Warszawski et al. (2014) explain such? (I found they said "process understanding and model development" for ISIMIP.)

**P3L28–32** Are there any relationship between ISI-MIP and BES-SIM? If they have, please explain. Else, please remove this sentence.

**P4L7** assess -> assessed

**2 Scenario selection** It is a little bit obvious that there is no description for bioenergy in SSP1×RCP2.6.

Regarding this, there is no information for SSP×RCP scenario matrix in this session. Therefore, I don't fully understand what is the rationale in the selection

**GMDD**

of scenario. I believe careful explanation in the scenario selection is helpful to communicate the results.

**P7L6–7** Please add the citations for all the biodiversity models here. This line is the first appearance of the models in the text.

**P7L4–11** Please add the citation in the manuscript, even though author remarked the citations.

What is "PREDICTS"? CO2 concentrations come from RCP? etc...

**Section 4** Please remark specifically how many species (taxonomic groups?) can simulate in each model.

**P7L24–25** Please suggest the definitions and units in each variable, if they have (e.g., species, amphibians, organisms).

**P8L6** What is the expert-based model?

**P8L7** Please add the brief explanation for BIOMOD2.

**P8L11–21** Please suggest the definitions and units in each variable.

**P8L27** What is species-area relationship? species abundance-area relationship?

**P8L32** "a hierarchical mixed-effects framework to model" -> just "hierarchical mixed-effects model"

**P8L33** "global database" <- Please add the citation.

**P9L20** Please clarify what ecosystem states are in DGVM. Also for functioning and habitat structure.

**5** Please add the units in each output (variables).

**P10L6** "was calculated at two scales: local or grid cell ($\alpha$)" <- Is this correct? The area of grid differs among different latitudinal bands. So, the scale is not unique when the grid cell is used.

**P10L10–11** Please describe the definitions of intactness in the text.

**P10L16–17** I'm strongly doubtful whether just one-year baseline is appropriate. But, I'm not sure how large variances in the year to year change are existing in such projected variables.

**P10L18–31** Please add the units for each outputs (e.g., kg-C/m2/year, kg-algae/L etc...).

**P11L9** What is "IPBES region"? Please show us the map.

**P11L17–19** Why? I'm not sure "the gap in climate input".

**P11L24** "different model parameterizations" <- I (and readers) cannot follow this meaning. Which parameters? Why they have the uncertainty range. How many simulations are used to get the quantiles of metrics?

**P11L25** Option 2 seems to assess the uncertainty just for BIOMOD model. This is not inter-comparison among BES-SIM models.

**Section 8** First paragraph seemed to be just repeatment of introduction. Please remove this paragraph.

**Section 8** In my opinion, the discussion is not essential in this paper, because of nothing results. Instead of discussion, please summarize uncovered topics in the current BES-SIM framework from the view to biodiversity (ESs) projection.

**P20L20** Please revise the author list in Settele et al.

**P33 Appendix1** Please remove "?" in "Three functional groups?" in Madingley model

---

## Author Comment (AC1) · 28 Sep 2018

Dear Referees,

We thank you once again for considering our manuscript at Geoscientific Model Development. We appreciate your positive and thorough review and valuable and constructive suggestions from the two referees. Based on these comments, we have revised the previous version of the manuscript and added necessary additional information to improve it. Please kindly find our detailed response with a revised manuscript in the supplements.

Please also note the supplement to this comment:
https://www.geosci-model-dev-discuss.net/gmd-2018-115/gmd-2018-115-AC1-
supplement.pdf

—————————————————————

---

## Author Response (AR1)

**Response to Reviewer's Comments**

We would like to thank the reviewers for the thorough review and comments on our manuscript. In the revised version, we did our best to incorporate them and we feel that the manuscript has greatly improved as a result. Please see the specific replies to the reviewer comments below.

**Anonymous Referee #1**

**Received and published: 30 August 2018**

The manuscript outlines a protocol for comparing models of the impacts of land-use and climate on biodiversity and ecosystem services. Intercomparisons of biodiversity and ecosystem services models and scenarios is much needed, especially for intergovernmental processes such as the IPBES, other policy processes, and conservation interventions. The manuscript outlines a process of comparing 16 spatially-explicit models for past (up to 900 AD), present (2015) and future (up to 2070) based on 3 scenarios (combinations of SSPs and RCPs) and the output variables that can be compared. Overall, biodiversity and ecosystem services model intercomparisons are much needed and this manuscript outlines a protocol for such intercomparisons for the first time.

**Thank you for your comments.**

Uncertainty is a critical part of models and model intercomparisons as acknowledged in the manuscript (section 7). The section on uncertainties, how uncertainty will be assessed within models and across models is too brief to be helpful. It would be interesting to better understand what the "comprehensive uncertainty analysis based on a variance partitioning approach" would involve. Furthermore, the text states that uncertainty of "the models of biodiversity" (P11) will be assessed, but there is no mention on how ecosystem services model outputs will be assessed for uncertainty. Both types of models will contain uncertainties that require assessment.

We appreciate this comment. In BES-SIM, uncertainties were reported by each modelling team with quantiles, both for biodiversity and ecosystem services where feasible (i.e. models that are able to incorporate uncertainty in their structure). Using the outputs provided, we explored intermodel uncertainty by calculating the mean and standard deviations across all models. The proposed variance partitioning approach is a forward-looking analysis that we aim to produce, but that was not part of the first iteration of BES-SIM. Therefore, we removed it from this section and updated the text accordingly. Please see page 13 (P13L16-24).

The section on other input data (section 3.3) should acknowledge the need for additional parameters within each model, in particular in ecosystem service models. For example, InVEST requires detailed information on parameters/look up tables to allocate the ecosystem service; Madingley has predator-prey relationships encoded. Will the default values be used for the intercomparisons or will models be

modified? Which versions of the models will be used? Some of this information is provided in Appendix 1, however more detail could be provided.

In this protocol manuscript, we documented for each model its key components and any particular modifications made for this exercise (Appendix 1 and Table S2). Given the vast amount of technical information for each model and this intermodel comparison, we tried to keep the level of technical details to the extent of readability of the manuscript. Further details on the parameterization of each model, including its default values, can be found in the key publications of each model listed in Appendix 1 of the manuscript and a further set of publications being prepared as a special issue in Global Change Biology detailing each model.

**Minor comments**

P2L27: ecosystems are a subset of biodiversity as defined by the Convention on Biological Diversity, therefore delete "ecosystems" here.

**Corrected. Please see page 2 (P2L29).**

P2L31-32: the statement that land-use change has immediate impacts on biodiversity and ecosystem services and impacts of climate change involves time lags is not correct. Both land use and climate changes can have immediate and lagged impacts. There is substantial evidence that climate change can have immediate impacts (e.g. Wellbergen et al. 2008 Proc Roy Soc B 275: 419-425) and land use impacts can be time lagged (e.g. McMichael et al. 2017. Ancient human disturbances may be skewing our understanding of Amazonian forests. PNAS 114: 522-527; Jakovac et al. 2016. Land use as a filter for species composition in Amazonian secondary forests. J. Veg. Sci. 27: 1104-1116; Graham et al. Graham et al. 2017. Implications of afforestation for bird communities: the importance of preceding land-use type. Biodiv. Cons. 26: 3051-3071).

We have used suggested references (Wellbergen et al., 2008; Graham et al., 2017) and revised the sentence as follows on page 3 (P2L32-35):

"Habitat and land-use changes, resulting from past, present and future human activities, as well as climate change, have both immediate and long term impacts on biodiversity and ecosystem services (Graham et al., 2017; Lehsten et al., 2015; Welbergen et al., 2008)."

P3L8 Ferrier et al. 2016 missing from reference list, maybe should be IPBES 2016?

We corrected the reference to IPBES 2016, see for example on page 3 (P3L31).

P5L21 Need to clarify the difference between RCP7.0 which was used for land-use projections and RCP6.0 which is used for other scenario production. Explain why SSP3/RCP7.0 was not used instead of the mixed SSP3/RCP6.0+RCP7.0 for land use. Furthermore, Table 5 does not show the use of RCP7.0 for

land use, it is shown as RCP6.0; check this is correct.

The land use dataset used, LUH2, was produced to be consistent with paired RCPs. However, and although the SSP3 is associated with RCP7.0 (SSP3xRCP7.0), currently, climate projections (i.e., time series of precipitation and temperature) are not available for RCP7.0. Therefore, we chose the closest RCP available, which was RCP6.0, and adapted the name of this SSPxRCP combination to SSP3xRCP6.0 to fit our exercise. We now make this clearer in the text on page 5 (P5L17-20):

"The SSP3 is associated with RCP7.0 (SSP3xRCP7.0); however, climate projections (i.e., time series of precipitation and temperature) are currently not available for RCP7.0. Therefore, we chose the closest RCP available, which was RCP6.0, for the standalone use of climate projections and chose SSP3xRCP6.0 for the land use projections from the LUH2. In this paper, we refer to this scenario as SSP3xRCP6.0."

P6L16: spell out ESM at first use.

ESM is spelled out at its first use on page 4 (P4L5).

P7L21: Table 2 shows 13 (and not 12) models requiring climate data. Which is correct, text or table?

Corrected. Please see page 8 (P8L6)

P10L12-15: reword this sentence, not comprehensible.

We have rewritten this sentence to increase clarity as follows on page 12 (P12L2-7):

"The habitat change (H) measures cell-wise changes in available habitat for the species. It is the changes in the suitable habitat extent of each species relative to a baseline, i.e., (Ei,t-Ei,t0)/Ei,t0, where Ei,t is the suitable habitat extent of species i at time t within the unit of analysis. It is reported by averaging across species occurring in each unit of analysis (grid cell, region, or globe), and is provided by the species-level models (i.e., AIM-biodiversity, InSiGHTS, MOL) (Table 4)."

P10L30: "units of the metrics" are not listed in Table 6. Update Table 6 with units, or reword text.

Thank you for noting this. We now state that units are in Table S3 of the Supplement on page 12 (P12L22-23).

P11L3: replace "Additional" with "Additionally"

Corrected. Please see page 12 (P12L26).

P11L4: Table 5 does not show the multiple time points from past to future, this information is provided in Appendix 1 first table. Note tables in Appendix 1 do not have legends or numbers.

Indeed, we now refer to Appendix 1. We have now numbered Appendix 1 table as Appendix 1 Table A1. Please see page 34.

P12L4-5: insert "to" before "the CBD and. . . " and before "other relevant stakeholders"

The paragraph that contained this sentence has been deleted in response to a comment of the second reviewer to avoid the repetition of information from the Introduction.

P14References: some references included "edited by" information for journal articles, e.g. Harfoot et al 2014b, Heinimann et al 2017. Check this is in line with the journal reference guidelines. Several references are submitted or in preparation, hence make it impossible to fully assess this manuscript.

Thanks for noting this. We have now corrected all of the references with editors according to the Copernicus reference style. We have also updated the publishing status of the four references in preparation or submission with a URL link to the preprint where available. Please see for example on page 16 (P16L2-4)

P23-all tables: check carefully throughout. CO2 and m2 should have subscript and superscript "2"s.

Thank you for noting this. We now have subscripts and superscripts correctly placed, see for example of  $CO_2$  in Table 1 on page 28.

P23Table1: Information on RCP6.0 is provided, however as RCP7.0 is used for land use projection, some info needs to be provided for RCP7.0 in this table or elsewhere. Information on climate policies is missing for RCP6.0. Table legend should read "Sources of land use and climate input data in BES-SIM" as other input data are used in all models (see later Tables).

Please see our reply to the reviewer's comment P5L21. Further, we added the missing information regarding "climate policies" in RCP6.0 in Table 1 on page 28. We have also modified the caption of Table 2 as suggested on page 29.

P26Table4: better explain "alpha and gamma metrics".

In Section 5 Output metrics, we now provide definitions to alpha ( $\alpha$ ) and gamma ( $\gamma$ ) metrics as follows on page 11 (P11L18-20):

"These metrics were calculated at two scales: local or grid cell ( $\alpha$  scale, i.e. the value of the metric within the smallest spatial unit of BES-SIM which is the grid cell) and regional or global ( $\gamma$  scale, i.e. the value of the metric for a set of grid cells comprising a region)."

P32: Scholes et al. 2005 reference is missing in reference list

We added Scholes and Bigg 2005 reference to the list. Thank you for your note.

P34: spell out PFTs at first use (and all other acronyms throughout, e.g. GIS on P36, etc.)

All acronyms are now spelled out at their first use in Appendix 1, see for example on pages 36 and 38.

P37: PCR-GLOBWB is missing from the list of acronyms

We have now added it to the list of acronyms in Appendix 2 on page 43.

P1-37: throughout the text reference is being made to Table S1, S2 etc. (e.g. P7L11), however no tables S1, S2 etc. are included.

We now added a sentence at the end of the manuscript on page 15 (P15L1-2) stating the existence of the Supplement.

**Anonymous Referee #2**

Received and published: 31 August 2018

**Dear authors**

The manuscript "A protocol for an intercomparison of biodiversity and ecosystem services models using harmonized land-use and climate scenarios" could be suitable for Geoscientific Model Development. This manuscript explains the framework of BESSIM (inter-comparison study). This model inter-comparison study tries to contribute not only to science communities including earth system modeling, climate science, and ecology, but also the other stakeholders such as policy maker. I fully agreed on the importance of this project to manage the impact on biodiversity and ecosystem services by anthropogenic activities. However, I feel the main manuscript is the lack of content to follow the overall picture of BES-SIM.

From the view to GMD paper, I cannot recommend acceptance for the publication in the current manuscript.

**Major comments**

Many concerns are as follow;

 For outside of the ecological area of expertise, the manuscript is not so helpful to understand BES-SIM. For example, there are many outputs related to the biodiversity, however, there are few definition and explanation of each output. At least, this journal's fields is not ecology. So, readers cannot follow even the unit of outputs provided in BES-SIM (e.g., abundance). They should be clearly described in the main manuscript.

Thank you for your comments. We have now expanded the definitions of each output metric to better guide non-specialists readers throughout the manuscript. Please see Section 4 and Section 5.

2. (I guess) To each SSP (RCP), the narratives (and interpretation) from the view to biodiversity and its social governance should be described in the scenario session. Table 1 provides such kind of information against each factor. The readers want to know these narratives in advance. For example, which one is business as usual to biodiversity? If there are no narratives in biodiversity (it means biodiversity sector is just passive against the other policies), please explain instead. Or, please make the narratives so as to follow the each SSP concept. (Sorry, I wrote this comment before reading the discussion. But, even after the reading discussion, I keep this comment. Also, I apologize that I cannot read Rosa et al. (2017) due to the non-license to read.)

The SSPs purposely excluded biodiversity in its narratives, except for SSP1 being an environmentally

friendly scenario; thereby potentially the best for biodiversity. We have now added couple of sentences to make it explicit as follows on page 5 (P5L18-21):

"The SSP scenarios excluded elements that have interaction effects with climate change except for SSP1, which focuses on environmental sustainability. Thus, SSPs describe futures where biodiversity is not affected by climate change to allow for the important estimation of the climate change impact on biodiversity (O'Neill et al., 2014)."

Individual comments

P2L17–19 Please suggest the combination (i.e., SSP1×RCP2.6?) here.

We have inserted the three scenarios combinations in the abstract on page 2 (P2L20).

P3L2 Please clarify "at high levels of climate change".

We have deleted this part from the sentence for clarity on page 3 (P3L4).

P3L5–6 Please suggest how to assess the impacts using the scenario? With empirical or mechanistic model?

We have now refined this by adding the following text on page 3 (P3L8-12):

"Models are used to quantify the biodiversity and ecosystem services impacts of different scenarios, based on climate and land-use projections from General Circulation Models (GCM) and Integrated Assessment Models (IAM) (Pereira et al., 2010)."

P3L6-8 Models are ... Meaning of this sentence is not clear.

We have now refined this by adding the following text on page 3 (P3L12-14):

"These models include empirical dose-response models, species-area relationship models, species distribution models and more mechanistic models such as trophic ecosystem models (Pereira et al., 2010; Akçakaya et al., 2016)."

P3L9 I'm not sure "cross-model harmonization".

We replaced the term with "intermodel comparison" on page 3 (P3L15). Thank you for your note.

P3L9–12 "addressing this issue" <Please add the citation.

We replaced the term "this issue" with "this gap" to make it explicit that we refer to the gap identified in the previous sentence. We also added citations (IPBES, 2016; Leadley et al., 2014) that mention this gap

and clarified the role of this manuscript in addressing this gap as follows on page 3 (P3L14-19):

"So far, each of these scenario exercises have been based on a single model or a small number of biodiversity and ecosystem service models, and intermodel comparison and uncertainty analysis have been limited (IPBES, 2016; Leadley et al., 2014). The Expert Group on Scenarios and Models of the Intergovernmental Science-Policy Platform on Biodiversity and Ecosystem Services (IPBES) is addressing this gap by carrying out a biodiversity and ecosystem services model intercomparison with harmonized scenarios, for which this paper lays out the protocol."

P3L21–23 I didn't find Ferrier et al. (2016) said the meaning of this sentence in the discussion. Is this a correct citation?

We corrected the reference to IPBES 2016, see for example on page 3 (P3L31).

P3L25–27 I cannot agree with the sentence "to improve the robustness and comprehensiveness". MIPs (at least without observation data) will not contribute to the improvement of the robustness. Did Warszawski et al. (2014) explain such? (I found they said "process understanding and model development" for ISIMIP.)

Thank you for your comments. We revised the sentence to correct the mention of the robustness, and make clear that MIPs allow for identification of uncertainties associated with scenarios and models. We also added a citation as follows on page 3 (P3L32-34):

"Model intercomparisons bring together different communities of practice for comparable and complementary modelling, in order to improve the comprehensiveness of the subject modelled, and to estimate uncertainties associated with scenarios and models (Frieler et al., 2015)."

P3L28–32 Are there any relationship between ISI-MIP and BES-SIM? If they have, please explain. Else, please remove this sentence.

We added the following sentence to the next paragraph to clarify the relationship between ISI-MIP and BES-SIM as follows on page 4 (P4L12-13):

"Whereas independent of the ISI-MIP, the BES-SIM has been inspired by ISI-MIP and other intercomparison projects and was delivered to address the needs of the global assessment of IPBES."

P4L7 assess -> assessed

Corrected. Please see page 4 (P4L18).

2 Scenario selection It is a little bit obvious that there is no description for bioenergy in SSP1×RCP2.6. Regarding this, there is no information for SSP×RCP scenario matrix in this session.

Therefore, I don't fully understand what is the rationale in the selection of scenario. I believe careful explanation in the scenario selection is helpful to communicate the results.

The SSP and RCP scenarios explore low and high levels of changes in land use and climate. Bioenergy aspects are not prescribed by the storylines in scenarios but it is an endogenous outcome of the models applied on the SSPs in combination with the RCPs. The bioenergy is fairly low in SSP1xRCP 2.6 (highest mitigation) with low emissions of RCP2.6 in a low energy intense and sustainable SSP1 world. It is fairly high in SSP3xRCP 6.0 (little mitigation) even with little mitigation as, for instance, other land based mitigation is not working due to governance failures. SSP5xRCP8.5 (no mitigation) has the lowest bioenergy mainly based on residues. We have now added this information in Table 1 (c) on page 28.

P7L6–7 Please add the citations for all the biodiversity models here. This line is the first appearance of the models in the text.

We have revised the Section 3.3 and Section 4 where the models are introduced, to have the references for the models to appear the first time they are mentioned. Thank you for noting this.

P7L4–11 Please add the citation in the manuscript, even though author remarked the citations.

We have revised the Section 3.3 and added citations in the manuscript.

**What is "PREDICTS"?**

PREDICTS (Projecting Responses of Ecological Diversity In Changing Terrestrial Systems) is the name of one of the biodiversity models included in the BES-SIM exercise. In reality, PREDICTS is a research project (http://www.predicts.org.uk/), in which they create a database of field data (collected from the literature) (Hudson et al. 2017; Hudson et al. 2016) and with which they developed their biodiversity model (Newbold et al., 2016; Purvis et al., 2018). In order to ease the read of the manuscript, which has a large number of acronyms, we kept the model names in acronyms in the text to prevent acronym cluttered sentences which are difficult to follow. The full names of the models can be found both in Appendix 1 and Appendix 2.

CO2 concentrations come from RCP? etc...

To maintain readability, we opted by keeping the text of the manuscript brief with a reference to the table where data sources can be found either in the table itself or in the key publications for each model.

Section 4 Please remark specifically how many species (taxonomic groups?) can simulate in each model.

We have now added the taxonomic groups and the number species modelled in each model in Section 4.

P7L24-25 Please suggest the definitions and units in each variable, if they have (e.g., species, amphibians,

organisms).

We have added definitions to the metrics, ecological concepts and methods used in modelling, and taxonomic groups and the number of species modelled in Section 4 to better guide the reader.

P8L6 What is the expert-based model?

We now describe expert-based models and maps as for general comment and clarified it in the sentence as follows on page 8 (P8L25-30):

"Both models rely on IUCN's expert-based range maps as a baseline, which are developed based on expert knowledge of the species habitat preferences and areas known to be absent (Fourcade, 2016). InSiGHTS and MOL used a hierarchical approach with two steps: first, a statistical model trained on current species ranges is used to assess future climate suitability within species ranges; second, a model detailing associations between species and habitat types based on expert opinions is used to assess the impacts of land-use in the climate suitable portion of the species range."

P8L7 Please add the brief explanation for BIOMOD2.

We added brief description of the BIOMOD2 model as follows on page 8-9 (P8L30-P9L2):

"BIOMOD2 is an R modelling package that runs up to nine different algorithms (e.g., random forests, logistic regression) of species distribution models using the same data and the same framework. BIOMOD2 included three taxonomic groups (amphibians, birds, mammals) (see section 7. Uncertainties)."

P8L11–21 Please suggest the definitions and units in each variable.

Thank you for your comment. We edited this paragraph with definitions and units where feasible to increase clarity. Please see the new text on page 9 (P9L4-17):

"Community-based models predict the assemblage of species using environmental data and assess changes in community composition through species presence and abundance (D'Amen et al., 2017). Output variables of community-based models include assemblage-level metrics such as the proportion of species persisting in a landscape, mean species abundances (number of individuals per species), and compositional similarity (pairwise comparison at the species level) relative to a baseline (typically corresponding to a pristine landscape).

Three models in BES-SIM – cSAR-iDiv (Martins and Pereira, 2017), cSAR-IIASA-ETH (Chaudhary et al., 2015), BILBI (Hoskins et al., submitted.; Ferrier et al., 2004, 2007) – rely on versions of the species-area relationship (SAR) to estimate the proportion of species persisting in human-modified habitats relative to native habitat (i.e., number of species in modified landscape divided by number of species in the native

habitat). In its classical form, the SAR describes the relationship between the area of native habitat and the number of species found within that area. The countryside SAR (cSAR) builds on the classic SAR but accounts for the differential use of both human-modified and native habitats by different functional species groups."

P8L27 What is species-area relationship? species abundance-area relationship?

As noted in the response to the previous comment, we have now added explanations on species area relationship and countryside species area relationship. For each ecological concept introduced and used in models in the manuscript, we now give brief description.

P8L32 "a hierarchical mixed-effects framework to model" -> just "hierarchical mixed effects model"

Corrected. Please see page 10 (P10L5).

P8L33 "global database" <Please add the citation.

We added sources (Hudson et al. 2017; Hudson et al. 2016) for the database as suggested on page 9 (P10L7). Thank you for your note.

P9L20 Please clarify what ecosystem states are in DGVM. Also for functioning and habitat structure.

We have added examples in the sentence as follows on page 10-11 (P10L30-P11L4). In addition, the details of the DGVMs included in BES-SIM are documented in Table S2 of the Supplement, and can also be found in the model specific publications referenced.

"DGVMs can project changes in future ecosystem state (e.g., type of plant functional trait (PFT), relative distribution of each PFT, biomass, height, leaf area index, water stress), ecosystem functioning (e.g., moderation of climate, processing/filtering of waste and toxicants, provision of food and medicines, modulation of productivity, decomposition, biogeochemical and nutrient flows, energy, matter, water), and habitat structure (i.e., amount, composition and arrangement of physical matter that describe an ecosystem within a defined location and time); however, DGVMs are limited in capturing species-level biodiversity change because vegetation is represented by a small number of plant functional types (PFTs) (Bellard et al., 2012; Thuiller et al., 2013)."

5 Please add the units in each output (variables).

We added units to output metrics mentioned in the section.

P10L6 "was calculated at two scales: local or grid cell ( $\alpha$ )" <Is this correct? The area of grid differs among different latitudinal bands. So, the scale is not unique when the grid cell is used.

All alpha ( $\alpha$ ) values are proportions calculated relative to an historical baseline in each cell. Therefore, they are not affected by variations of cell area across latitude. The exception is N $\alpha$ , where absolute changes are calculated relative to an historical baseline. Still, the variations in cell area have limited impact on these absolute changes which are mostly influenced by land-use and climate changes and by the latitudinal gradient in species richness.

P10L10–11 Please describe the definitions of intactness in the text.

We added the definition of abundances and intactness in the text as follows on page 11-12 (P11L30-P12L2):

"The abundance-based intactness (I) measures the mean species abundance in the current community relative to the abundances in a pristine community. This metric is available only for two community-based models, i.e., GLOBIO (where intactness is estimated as the arithmetic mean of the abundance ratios of the individual species, whereby ratios >1 are set to 1), and PREDICTS (where intactness is estimated as the ratios of the sum of species abundances)."

P10L16–17 I'm strongly doubtful whether just one-year baseline is appropriate. But, I'm not sure how large variances in the year to year change are existing in such projected variables.

Biodiversity models in our analysis respond to land-use change and climate change. Land-use changes are relatively smooth over time and there are no large year to year fluctuations. Climate can indeed exhibit large scale annual fluctuations, particularly at the local level, but for historical projections for the year 1900 the biodiversity models assumed no climate impacts on biodiversity metrics. So we believe a single baseline year of 1900 is appropriate here.

P10L18–31 Please add the units for each outputs (e.g., kg-C/m2/year, kg-algae/L etc...).

We added units for each output as suggested. Please see page 12 (P12L10-24).

P11L9 What is "IPBES region"? Please show us the map.

We added references to a map and a dataset showing IPBES regions and subregions (Brooks et al. 2016; UNEP-WCMC, 2015). Please see page 13 (P13L1-2).

P11L17-19 Why? I'm not sure "the gap in climate input".

We revised the sentences as follows on page 13 (P13L9-14):

"When backcasting to 1900, for the models that required continuous climate time-series, random years in the period 1951 to 1960 from the ISIMIP 2a IPSL climate dataset were used to fill the data missing for years 1901 to 1950. Models that used multi-decadal climate averages (i.e., InSiGHTS, BILBI) assumed no climate impacts for 1900."

P11L24 "different model parameterizations" <I (and readers) cannot follow this meaning. Which parameters? Why they have the uncertainty range. How many simulations are used to get the quantiles of metrics?

We changed it to "based on the parameterizations specific to each model" for clarity on page 13 (P13L18). The way in which each model produced the quantiles depends on the model's internal structure, but in general these arise from varying internal model parameters. For instance, in the cSARiDiv model, the uncertainty arises from performing a Monte Carlo/bootstrapping on the affinity values (i.e., sample 100 different values), which leads to slightly different model outputs (i.e., 100 maps of projected extinctions per year). The quantiles are then derived from the distribution of these 100 model outputs (computed using different affinities and the same land use). The details of how uncertainty was provided by each of the models can be found in either their key publications (listed in Appendix 1) or in the Table S2 of the Supplement if this was an improvement made specifically for BES-SIM (e.g., as is the case of cSAR-iDiv).

P11L25 Option 2 seems to assess the uncertainty just for BIOMOD model. This is not inter-comparison among BES-SIM models.

The reviewer is correct. We revised the text as follows to differentiate the two elements of uncertainty analysis – one with quantiles provided by each model and their intercomparison, and another one specifically to climate projections by BIOMOD2, which is independent of the intermodel comparison. We revised the text as follows on page 13 (P13L16-24):

"Reporting uncertainty is a critical component of model intercomparison exercises (IPBES, 2016). Within BES-SIM, uncertainties were explored by each model reporting the mean values of its metrics, and where possible the 25th, 50th, and 75th percentiles based on the parameterizations set specific to each model, which can be found in each model's key manuscripts describing the modelling methods; and when combining the data provided by the different models, the average and the standard deviation of the common metrics were calculated (e.g., intermodel average and standard deviation of  $P\gamma$ ). In a parallel exercise to inform BES-SIM, the BIOMOD2 model was used in assessing the uncertainty in modelling changes in species ranges arising from using different RCP scenarios, different GCMs, a suite of species distribution modelling algorithms (e.g., random forest, logistic regression) and different species dispersal hypotheses."

Section 8 First paragraph seemed to be just repeatment of introduction. Please remove this paragraph.

**We removed the paragraph as suggested.**

Section 8 In my opinion, the discussion is not essential in this paper, because of nothing results. Instead of

discussion, please summarize uncovered topics in the current BES-SIM framework from the view to biodiversity (ESs) projection.

This section is now called "Conclusion" to better reflect the manuscript. Uncovered topics in this BES-SIM framework will be published in a separate manuscript; thus, we kept the remaining text.

P20L20 Please revise the author list in Settele et al.

We added the full author list for the Settele et al. reference.

P33 Appendix1 Please remove "?" in "Three functional groups?" in Madingley model

We corrected this typo.

**A protocol for an intercomparison of biodiversity and ecosystem services models using harmonized land-use and climate scenarios**

HyeJin Kim1,2, Isabel M.D. Rosa1,2, Rob Alkemade3,4, Paul Leadley5, George Hurtt6, Alexander Popp7, Detlef P van Vuuren8, Peter Anthoni9, Almut Arneth9, Daniele Baisero10, Emma Caton11, Rebecca Chaplin-Kramer12, Louise Chini6, Adriana De Palma11, Fulvio Di Fulvio13, Moreno Di Marco14, Felipe Espinoza11, Simon Ferrier15, Shinichiro 5 Fujimori16, Ricardo E. Gonzalez18, Maya Gueguen29, Carlos Guerra1,2, Mike Harfoot19, Thomas D. Harwood15, Tomoko Hasegawa17, Vanessa Haverd20, Petr Havlík13, Stefanie Hellweg21, Samantha L. L. Hill11,19, Akiko Hirata22, Andrew J. Hoskins15, Jan H. Janse3,23, Walter Jetz24, Justin A. Johnson25, Andreas Krause9, David Leclère13, Ines S.

- Martins1,2, Tetsuya Matsui22, Cory Merow24, Michael Obersteiner13, Haruka Ohashi22, Benjamin Poulter26, Andy 10 Purvis11,27, Benjamin Quesada9,28, Carlo Rondinini10, Aafke M. Schipper3,2829, Richard Sharp12, Kiyoshi Takahashi17, Wilfried Thuiller29Thuiller30, Nicolas Titeux1,3031, Piero Visconti31Visconti32,3233, Christopher Ware15, Florian Wolf1,2, Henrique M. Pereira1,2,33\_34

[revised manuscript text omitted]